# Satb1 integrates DNA binding site geometry and torsional stress to differentially target nucleosome-dense regions

Rajarshi P. Ghosh[1,2,3,4,14], Quanming Shi[1,2,3,4,14], Linfeng Yang[1,2,3,4], Michael P. Reddick[1,2,3,4,5], Tatiana Nikitina[6], Victor B. Zhurkin[6], Polly Fordyce [1,3,7,8], Timothy J. Stasevich[9], Howard Y. Chang [7,10,11,12], William J. Greenleaf [7,13] & Jan T. Liphardt [1,2,3,4]

The Satb1 genome organizer regulates multiple cellular and developmental processes. It is not yet clear how Satb1 selects different sets of targets throughout the genome. Here we have used live-cell single molecule imaging and deep sequencing to assess determinants of Satb1 binding-site selectivity. We have found that Satb1 preferentially targets nucleosome-dense regions and can directly bind consensus motifs within nucleosomes. Some genomic regions harbor multiple, regularly spaced Satb1 binding motifs (typical separation ~1 turn of the DNA helix) characterized by highly cooperative binding. The Satb1 homeodomain is dispensable for high affinity binding but is essential for specificity. Finally, we find that Satb1-DNA interactions are mechanosensitive. Increasing negative torsional stress in DNA enhances Satb1 binding and Satb1 stabilizes base unpairing regions against melting by molecular machines. The ability of Satb1 to control diverse biological programs may reflect its ability to combinatorially use multiple site selection criteria.

[1] Bioengineering, Stanford University, Stanford, CA 94305, USA. [2] BioX Institute, Stanford University, Stanford, CA 94305, USA. [3] ChEM-H, Stanford University, Stanford, CA 94305, USA. [4] Cell Biology Division, Stanford Cancer Institute, Stanford, CA 94305, USA. [5] Chemical Engineering, Stanford University, Stanford, CA 94305, USA. [6] Laboratory of Cell Biology, Center for Cancer Research, National Cancer Institute, National Institutes of Health, Bethesda, MD 20892, USA. [7] Department of Genetics, Stanford University, Stanford, CA 94305, USA. [8] Chan Zuckerberg Biohub, San Francisco, CA 94158, USA. [9] Department of Biochemistry and Molecular Biology and the Institute for Genome Architecture and Function, Colorado State University, Fort Collins, CO, USA. [10] Center for Personal Dynamic Regulomes, Stanford University, Stanford, CA 94305, USA. [11] Program in Epithelial Biology, Stanford University School of Medicine, Stanford, CA 94305, USA. [12] Howard Hughes Medical Institute, Stanford University, Stanford, CA, USA. [13] Department of Applied Physics, Stanford University, Stanford, United States. [14] These authors contributed equally: Rajarshi P. Ghosh, Quanming Shi. Correspondence and requests for materials should be addressed to J.T.L. (email: jan.liphardt@stanford.edu)

At the core of cellular information processing is the ability of transcription factors (TFs) to bind subsets of genomic targets selectively. Proteins with multiple DNA Binding Domains (DBD) can combinatorially engage multiple distinct core DNA consensus motifs[1,2]. The DNA backbone and major and minor groove shape constitute a second evolutionarily conserved constraint[3] recognized through indirect readout[4] or shape readout[5] mechanisms. Recently it has been suggested that dinucleotide features are sufficient to reliably predict DNA shape parameters[6]. Transcription factors have also been shown to streamline their binding choices based on the shape of regions flanking the core motif[7]. Beyond sequence and shape, geometric constraints such as motif-to-motif spacing, orientation, and local density of binding sites influence cooperative TF binding[8,9]. Finally, TF binding is further modulated by chromatin accessibility and nucleosome occupancy[10,11] which in turn are affected by DNA torsional stress and deformability[12–14]. Our goal was to explore how the interplay of these parameters defines the genome-wide distribution of the chromatin loopscape regulator[15] Special AT-Rich Sequence Binding Protein 1 (Satb1).

Satb1 is a dimeric/tetrameric[16] transcription factor with multiple DNA binding domains, namely CUT1, CUT2 and a C-terminal homeodomain (HD). Satb1 has been implicated in diverse cellular processes including epidermal differentiation[17], breast cancer metastasis[18], thymocyte development[19], Th2 cell activation and cytokine production[20], cortical development[21], X-chromosome inactivation[22], and embryonic stem cell differentiation[23]. Satb1 has also been deemed to be a genome organizer responsible for rapid phenotypic transitions[15]. For instance, 3C techniques applied to T cells suggest that Satb1 mediates transcriptional and epigenetic changes at target loci[20]. In the same vein, Satb1 is instrumental in establishing a Foxp3+ regulatory T cell ($T_{reg}$)-specific lineage by defining a $T_{reg}$ cell-specific super enhancer landscape[24].

Despite an abundance of in vitro biochemical and biophysical data, there is no consensus on Satb1's binding mechanism(s). In vitro and affinity-based pull-down experiments indicate that Satb1 binds A/T rich motifs with high base-unpairing potential (Base Unpairing Regions: BUR)[25,26], which have a propensity to unpair under torsional stress[27,28]. Chemical interference assays suggest that Satb1 binds along the minor groove of DNA with virtually no contact with the bases[25,29]. SELEX experiments suggest that the Satb1 homodimer binds an inverted AT-rich palindromic repeat along the minor groove[29]. Unlike other HD proteins, Satb1's HD does not independently bind an HD consensus sequence but appears to increase binding specificity towards BURs in vitro[26]. It has also been postulated that the spacing of the half-sites is critical for binding as a dimer. Finally, solution biophysical assays suggest that the sequence specificity is due to binding by CUT1 along the major groove[30,31] but does not require an AT-rich inverted palindromic repeat[31]. The disparate nature of these findings could reflect over-simplified in vitro biochemical settings (which exclude the native genomic environment and/or nucleosomal context) or the use of a narrow subset of DNA substrates with selection bias.

To elucidate how Satb1 binds its targets, we have combined live cell imaging (spatiotemporal FRAP[32], image correlation spectroscopy[33], and single molecule tracking[34,35]), genomics (ChIP-seq[36], ATAC-seq[37], ChIP-ATAC-seq, TMP-seq[13]) and in vitro nucleosome binding assays. Our work shows that Satb1 binds transposase-inaccessible, nucleosome-dense regions in chromatin and contacts motifs embedded in nucleosomal core sequences, both in vitro and in vivo. Satb1 binding shows repeated binding and unbinding to a small number of spatially proximal chromatin interaction hotspots. Using deep sequencing we show that of Satb1 binding sites exist in clusters in the genome, characterized by highly cooperative binding. Satb1 prefers regions of the DNA with multiple consensus motifs spaced ~10 bp apart. The HD is crucial for site-specificity, as it helps to home in on motifs with flanks having enhanced negative propeller twist (PT) and higher AT content than the genome average. Finally, Satb1 binds more efficiently to target sites under highly negative torsional stress and stabilizes BURs against helix destabilization, as evidenced by the abolition of psoralen crosslinking from binding sites in the presence of Satb1.

## Results

**Satb1 distribution reveals chromatin interaction hotspots.** In thymocytes, the pool of nuclear Satb1 appears to be concentrated in extended tendrils which may be important for thymocyte development[15]. To investigate whether the Satb1 molecules in these extended regions are organized into a static structure of some kind or if these tendrils represent an excluded volume of dynamically exchanging Satb1, we tagged native Satb1 with eGFP using a CRISPR/CAS9 knockin strategy in an immortalized CD4 + CD8 + thymocyte cell line (VL3-3M2, Supplementary Fig. 1a, b). Transient co-expression of histone H2B-mCherry revealed an inverse spatial correlation of Satb1 and dense heterochromatin, confirming previous immunofluorescence studies[20] (Fig. 1a, b), even though the levels of Satb1 in heterochromatin were clearly detectable (Fig. 1a, b).

We used image correlation spectroscopy (ICS) to measure the dynamics of Satb1-eGFP in the euchromatin, as well as at the euchromatin/heterochromatin (EC/HC) boundary. Figure 1c shows a carpet of pixel intensity along a 3.2 μm line bisecting a Satb1 tendril versus time. Temporal autocorrelation analysis[33] of intensities of each pixel along the bisecting line yielded diffusion coefficients of Satb1 specific to that pixel (Fig. 1d). The diffusion coefficients obtained from the average of all fits showed that Satb1 is highly dynamic in EC ($5.1 \pm 1.8 \, \mu m^2 \, s^{-1}$) and to a lesser extent at the EC/HC boundary ($1.7 \pm 1.1 \, \mu m^2 \, s^{-1}$). Therefore, the majority of the Satb1 molecules are highly dynamic albeit with varying ability to access different regions of the nucleus, raising the possibility that Satb1 tendrils are not static structures in their own right, but reflect cell-type specific features of the thymocyte heterochromatin.

We then used HiLo TIRF microscopy[38] to localize single, chromatin-bound Satb1 molecules. To set a baseline and assess how Satb1 binds to chromatin, we ectopically expressed a C terminal eGFP fusion of Satb1 (inducible via cumate control) in the MCF10A breast epithelial cell line. MCF10A lacks native Satb1. In this experimental geometry, the Satb1-eGFP will bind a genome that is unaffected by architectural changes and epigenetic modifications resulting from chronic Satb1 exposure. Unlike thymocytes which exhibit a specialized nuclear architecture, the Satb1 distribution in the MCF10A did not show marked depletion in heterochromatin (Fig. 1e, Supplementary Fig. 1c). The background (leaky) zero-cumate expression yielded a molecule observation density of ~30 per nucleus/frame in HILO TIRF imaging[38], which proved suitable for single molecule tracking.

Localization of single Satb1 molecules (Fig. 1f) revealed a wide spatial variation in signal density, ranging from regions with only a few localizations per $\mu m^2$ to hotspots with dozens to hundreds of localizations per $\mu m^2$ (Fig. 1f, zoom). In the MCF10A cells, we did not observe higher-order structural features of Satb1 (such as tendrils or droplets), and there were no apparent sub-nuclear spatial preferences (Fig. 1f, Supplementary Fig. 1c). The substantial (100-fold) spatial variation of Satb1 localizations could reflect heterogeneity in the distribution of Satb1 binding sites throughout the genome and/or slow (but cooperative) binding to random subsets of sites.

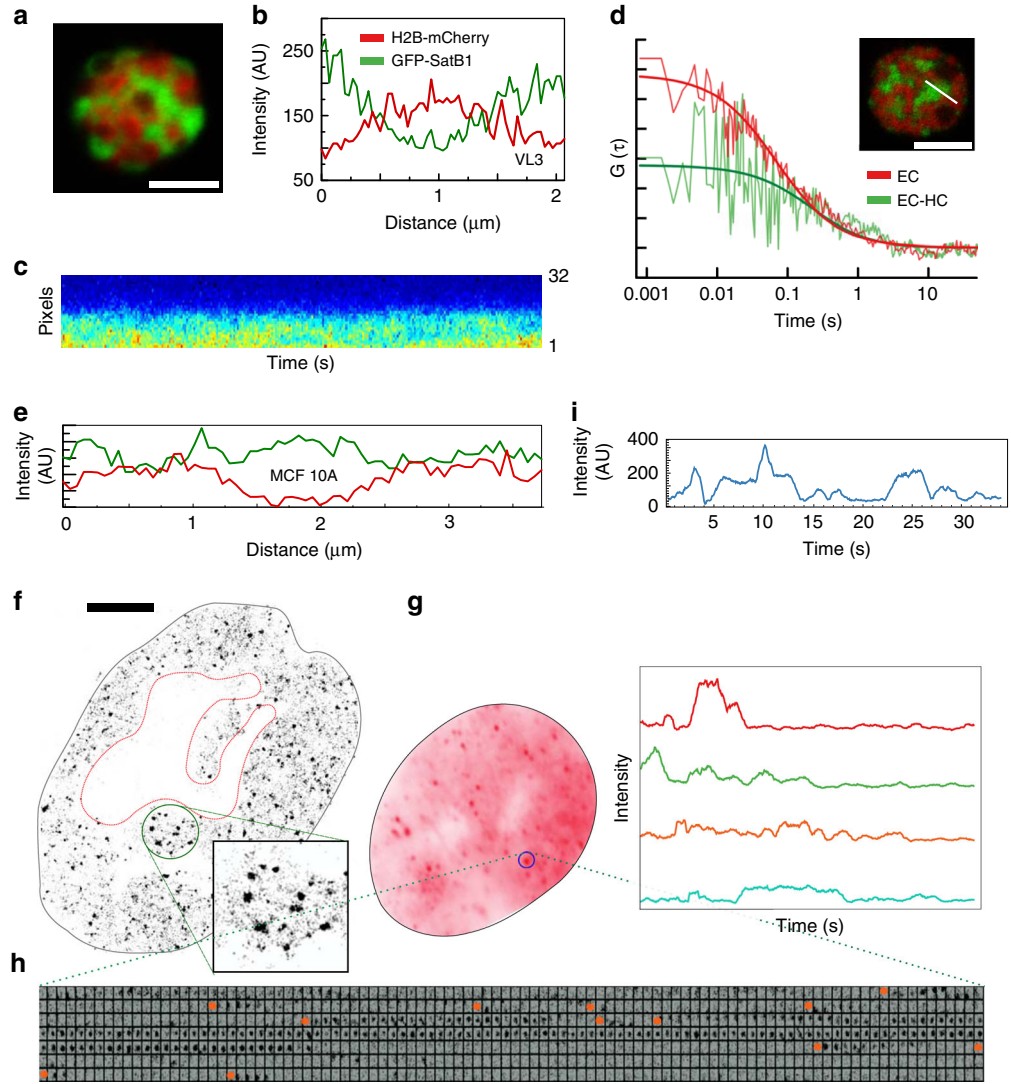

**Fig. 1** High resolution spatiotemporal analysis of SatB1 binding. **a** A VL3 3M2 thymocyte Satb1-eGFP knockin cell line co-expressing core histone H2B-mCherry shows the classic cage pattern. **b** Satb1-eGFP and H2B-mCherry profile along a line bisecting the thymocyte nucleus. **c, d** A 3.2 μm-line bisecting a Satb1 cage and an adjacent heterochromatin is scanned repeatedly with a pixel dwell time of 6.3 μs (32 pixels) and a line time of 0.47 ms. **c** A fluorescence intensity carpet generated by repeated scanning over time where the pixels are along y axis and time is along x axis. **d** The autocorrelation function is calculated pixel-by-pixel along the scanned line. The two autocorrelation functions shown are for one-pixel columns that reside either in the cage or at the cage-heterochromatin boundary. **e** Exogenous Satb1 distribution in MCF-10A cells does not show any stark difference between euchromatic and heterochromatic compartments. **f** Super-resolution image of Satb1 in the MCF-10A nucleus reconstructed from all localizations obtained over ~1000 frames. **g** Properties of the Satb1 interaction hotspots. The summed intensity of 700 frames (spanning 35 s) for a cell expressing eGFP fusion of native Satb1. Note discrete spot like structures scattered throughout the nucleus. **h** A 7 × 100 matrix of one region of interest (marked by circle in **f**) showing intensity changes over time. **i**. Intensity traces over time of a few chromatin interaction hot spots. Scale bar is 5 μm. SMT analysis including raw tracks are included in the Source Data file

To investigate the origins of Satb1's binding heterogeneity we studied μm² sized chromatin regions and asked whether Satb1 occupied some regions more frequently than others. When we summed the intensity data from the entire experiment, we noted bright spots that stood out from their local background (Fig. 1g). These regions were characterized by the repeated arrival, capture, and departure of Satb1 molecules (Fig. 1h-i). Next, we employed several deep sequencing tools to define the genomic sequence, nucleosome occupancy, and torsional state of the genomic DNA in regions preferred by Satb1.

**Satb1 has one dominant consensus motif.** To explore the genome-wide distribution of Satb1, we performed chromatin immuno-precipitation followed by deep sequencing (ChIP-seq) in VL3-3M2 and MCF10A cells. Although absolute level of Satb1 expression was ~7 fold less in VL3-3M2 cells compared to MCF10A cells, concentrations of Satb1 per unit nuclear volume were nearly identical owing to much smaller nuclear volume of VL3-3M2 (Supplementary Fig. 2a-e). The ChIP-seq data were highly replicable (Pearson correlation coefficient for ChIP-seq duplicates >0.995, Supplementary Fig. 3). Figure 2a, b shows a heat map centered on the Satb1 binding peaks and sorted by peak intensities in MCF10A cells. To validate whether Satb1 binding sites preferentially localized to A/T rich regions, we calculated the A/T percentage in the same symmetric 4 kb window as used for calculating ChIP-seq intensities. As expected, the A/T percentage heat-map was strongly correlated with ChIP-seq signal (Fig. 2b, c;

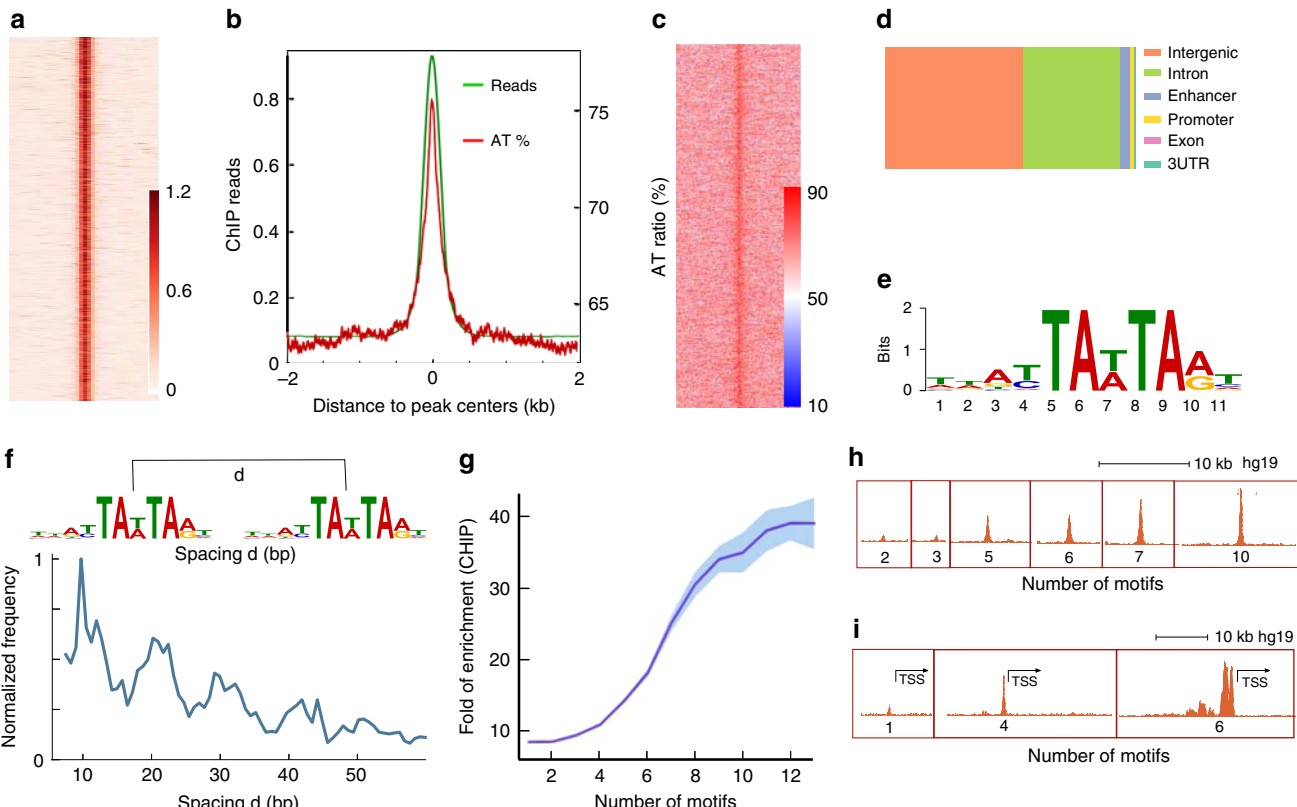

**Fig. 2** Genome wide binding of Satb1 reveals clustered distribution and cooperative binding. **a** Heatmap of ChIP-seq signals (read pileups) along a 4 kb window centered on binding peaks. **b** Overlay of average ChIP-seq signals (green) and AT percentage (red) along the same 4Kb window as in **a**. **c** Heatmap of AT percentage along the 4 kb window as in **a**. **d** Relative abundance of Satb1 binding sites in different genomic categories. **e** Identification of a Satb1 consensus motif in MCF10A using MEME. **f** Frequency distribution of the spacing between Satb1 bound motif pairs in a 400 bp window, showing ~ 10 bp periodicity. **g** Satb1 ChIP-seq shows cooperative binding (mean ChIP-seq signal versus number of motifs per binding site reveals a sigmoidal binding curve with a Hill coefficient of 5.3. **h i**. Representative examples of ChIP-seq signal strength classified by motif density per binding site in gene body (**h**) and proximal to TSS (**i**). All error envelopes represent standard error of mean. Source data are provided as a Source Data file

Supplementary Fig. 4a-c). A genome-wide scan for Satb1 targets revealed ~22,000 non-redundant binding sites in MCF10A and ~4000 sites in VL3-3M2, mostly located in intergenic and intronic regions (Fig. 2d, Supplementary Fig. 4d). To identify sequence motifs bound by Satb1, we performed de novo motif discovery using MEME[39] on a 150 bp DNA sequence centered on each ChIP-seq peak. For MCF10A, 91% of all binding sites were satisfied by a single consensus motif (Fig. 2e). 100% of all VL3-3M2-associated binding sites were satisfied by the same consensus motif (Supplementary Fig. 4e) whereas 22% of these sites were also satisfied by a second closely related consensus (Supplementary Fig. 4f). To test if the A/T rich inverted palindromes predicted by SELEX[29] constituted a subset of the major motif class in MCF10A (Fig. 2e), we searched for the two most common motifs predicted by SELEX[29]. We found only 192 occurrences of TATTAGTAATAA and 64 occurrences of TATTAGTAATAC out of 22,829 peaks, underlining the limitations of previous in vitro SELEX binding data in determining Satb1 consensus.

**Periodicity and clustering characterize Satb1 binding sites**. The most visually prominent feature of Satb1's distribution in MCF10A cells are the chromatin interaction hotspots. Do these hotspots correspond to regions of the chromatin that are highly enriched for Satb1 binding motifs? To gauge whether Satb1 binding sites are spatially clustered within the genome, we first detected all instances with two Satb1 motifs within a 400 bp window centered on a ChIP-seq peak (Fig. 2f). Further analysis revealed that these dual sites tended to be separated by ~10 bps

and that some regions had four or more such sites with ~10 bp spacing (Fig. 2f). A similar analysis for 8 other highly abundant transcription factors including several pioneer factors showed no such periodicity in motif spacing (Supplementary Fig. 5a), suggesting that ~10 bp inter-motif spacing is a unique structural constraint recognized by Satb1. Satb1 has been shown to bind well to BUR repeats in vitro and it has been speculated that the dimeric/tetrameric organization of Satb1 helps to stabilize its interaction with DNA through a monkey bar geometry over long[40], as well as short[31] distances.

To assess whether motif density affected binding strength, we plotted peak intensities against motif density in a 400 bp window centered on the ChIP-seq peak. Remarkably, the ChIP-seq signal strength increased cooperatively with motif density, with a notable Hill coefficient of 5.3 (Fig. 2g). A similar analysis of motif density versus ChIP-seq signal did not reveal cooperative binding for eight other transcription factors (Supplementary Fig. 5b). Figure 2h shows several examples of signal strengths at different motif densities. For Satb1, higher motif density correlated with higher signal strength, including regions surrounding the transcriptional start sites (TSS) (Fig. 2h, i). These findings indicate that motif density is an essential criterion in defining Satb1 site selectivity through cooperative recruitment of Satb1 molecules.

**Satb1 binds to nucleosome dense regions**. A genome-wide search of the Satb1 sequence consensus in MCF10A cells revealed ~2.6 million perfect sequence matches. Yet in vivo, Satb1 only

binds a tiny fraction (~0.8%) of these potential targets. We asked whether nucleosomes act as barriers to Satb1 binding since the ENCODE consortium has shown that most transcription factors bind to highly accessible (nucleosome-free) DNA regions[10,11]. We used ATAC-seq[37] to relate Satb1 binding sites to genome accessibility in MCF10A cells. For both native MCF10A cells and engineered cells expressing exogenous Satb1, the ATAC-seq fragment length distribution was enriched in nucleosome-free regions and within a dominant mono-nucleosome peak (Supplementary Fig. 6a). As expected, the general ATAC signal was highly enriched near transcriptional start sites (TSS, Supplementary Fig. 6b, c). Also, as expected, positive control data for CTCF binding (which prefers nucleosome-free regions) strongly overlapped with high ATAC signals (Fig. 3a). In sharp contrast, the bulk of the Satb1 target sites fell in transposase-inaccessible regions in native MCF10A cells (~96%) lacking Satb1 expression (Supplementary Fig. 7a), in MCF10A cells stably expressing Satb1 (~95%) (Fig. 3b), as well as native VL3-3M2 cells (~95%) (Supplementary Fig. 7b). Compared to Satb1 targets in transposase-accessible regions, a significantly higher fraction of Satb1 bound sites in transposase-inaccessible regions localized to promoters and enhancers (Fig. 3b). For Satb1 binding sites located in both inaccessible and accessible chromatin, the ATAC signal was visibly depleted (Fig. 3c, d), suggesting that Satb1 could be binding directly to nucleosomes. Satb1 binding sites in VL3-3M2 showed a very similar accessibility profile (Supplementary Fig. 7b).

**Satb1 binds to sequences embedded in the nucleosomal core**. The capacity to bind nucleosome-dense regions is a hallmark of pioneer factors[41]. Satb1 has been shown to have pioneer activity in Treg cells[24]. Satb1's preference for inaccessible chromatin made it inherently difficult to determine nucleosome positions at target sites due to low ATAC signal (Fig. 3b-d). We enriched sequencing reads that could be mapped to less accessible regions by performing ATAC-seq on Satb1 ChIP-enriched chromatin, and by removing reads shorter than one nucleosome length. This approach markedly increased the number of binding sites for which we could map nucleosome position[42]. The distribution of distances between the Satb1 binding center to the nearest nucleosome dyad showed that the bulk of Satb1 binding sites reside within the nucleosomal core (Fig. 3e, f). This pattern is opposite to CTCF, which binds to inter-nucleosomal linker regions (Fig. 3g)[37]. It has been suggested that transient unwrapping of core nucleosomal DNA allows pioneer factors to bind nucleosomes. To test whether Satb1 can bind nucleosomes directly by accessing target sites located in core nucleosomal sequences, we generated a modified version of the strong nucleosomal positioning sequence 601[43], where minor-groove outer facing nucleotides at three different superhelical locations (SHL3, SHL4, SHL6) were modified to generate three Satb1 consensus sites (601 A). Recombinant Satb1 robustly bound 601 A mono-nucleosomes (Fig. 3h), confirming the genomics data and establishing that Satb1 is capable of binding target sites inside the nucleosomal core sequence even in the absence of remodelers. While these findings strongly support Satb1's candidacy as a pioneer factor, they do not explain Satb1's selective binding to a small subset of putative target sites.

**Flanks with higher negative propeller twist promote binding**. To assess whether the shape of flanking sequences affects Satb1 selectivity, we compared the occupied to the non-occupied consensus Satb1 motifs. For each motif, we retrieved 100 bps on either side of the motif and evaluated its DNA shape[44]. The sequences were further categorized into nucleosome-free regions

(based on nucleosome occupancy analysis) and accessible or inaccessible chromatin (based on ATAC seq). Although nucleosome-free regions represent the least abundant class of Satb1 binding sites, this none the less was best suited for assessing the role of DNA shape on site selectivity, since these sites are free of structural distortions imposed by the nucleosomal histone scaffold[45]. The minor groove width (MGW), helical twist (HelT), and Roll varied only slightly with Satb1 occupancy, but bound motifs had higher negative propeller twist (PT) in their flanking sequences (Fig. 3i, j). Propeller twist may be a proxy for DNA flexibility[46]. All three sequence categories followed this pattern (Supplementary Fig. 8a, b) with the signal being most robust in the nucleosome-free regions (Fig. 3i, j). The similarity in shape parameters for sequences flanking Satb1 binding sites in both inaccessible and accessible chromatin (Supplementary Fig. 8a, b) is most likely due to nucleosome-imposed shape constraints on embedded sequence motifs and flanks. Indeed Nucleo-ATAC profiling of Satb1 binding sites in transposase accessible region showed preferential distribution inside nucleosomal core (Supplementary Fig. 9) similar to transposase inaccessible region (Fig. 3f). Because propeller twist strongly depends on the AT-content[46], we compared this parameter for the Satb1-bound and unbound targets. Satb1 bound sites showed higher A/T content in their motif flanks (75%) than the unbound ones (67%) (Fig. 3k, Supplementary Fig. 10). This suggests that A/T content and Propeller twist of motif flanks are important determinants of Satb1 site selectivity. Satb1 consensus sites (Fig. 2e), located preferentially in nucleosomal core sequences, have multiple TA base steps, which are flexible and known to have a stronger propeller twisting than other relatively rigid T/A dinucleotides[45,46]. This coupled with the frequent 10 bp periodic recurrence of Satb1 consensus sites (Fig. 2f) raises the intriguing possibility that the periodic enhancement of propeller twist at Satb1 binding site clusters may constitute an important selective filter. Thus, the enhanced propeller twisting in sequences flanking nucleosome-embedded consensus motif is more likely related to the unique geometry of TA base steps rather than a simple function of total A/T content.

**Satb1 stabilizes BURs against unwinding**. Classic Satb1 target sites such as the immunoglobulin heavy chain (IgH) enhancer region have been shown to melt over extended regions under torsional stress[26,28]. Torsion can be generated by replicating and transcribing polymerases, nucleosome assembly, and remodeling, as well as chromatin condensation by ATP dependent Condensins[12,47]. Topoisomerases maintain a torsionally relaxed genome by relieving torsional strain in chromatin that can build up during transcription and chromatin remodeling[12]. To trap the genome in a torsionally strained state we inhibited topoisomerase I (Topo I) and topoisomerase II (Topo II) in MCF10A cells for a short time with Camptothecin (CPT) and ICRF-193, respectively, and we then carried out ChIP-seq of Satb1 bound chromatin.

Upon treatment with ICRF-193, there was a visible increase in degree of binding for most binding sites (Fig. 4a, Supplementary Fig. 11). Camptothecin did not affect binding to low strength sites but increased binding to high strength sites (Fig. 4a, Supplementary Fig. 11). This increase in binding strength is further evident in Fig. 4b, which shows that Satb1 binds slightly more cooperatively upon ICRF-193, as well as CPT treatment (Fig. 4c). To delineate the torsional state of Satb1 binding sites, we carried out TMP-seq[13] on WT MCF10A cells and MCF10A cells stably expressing Satb1-eGFP. We also collected TMP-seq data from both of these cell types upon treatment with the topoisomerase inhibitors CPT and ICRF-193. Psoralen-crosslinking efficiency at Satb1 binding sites in native MCF10A cells diminished markedly

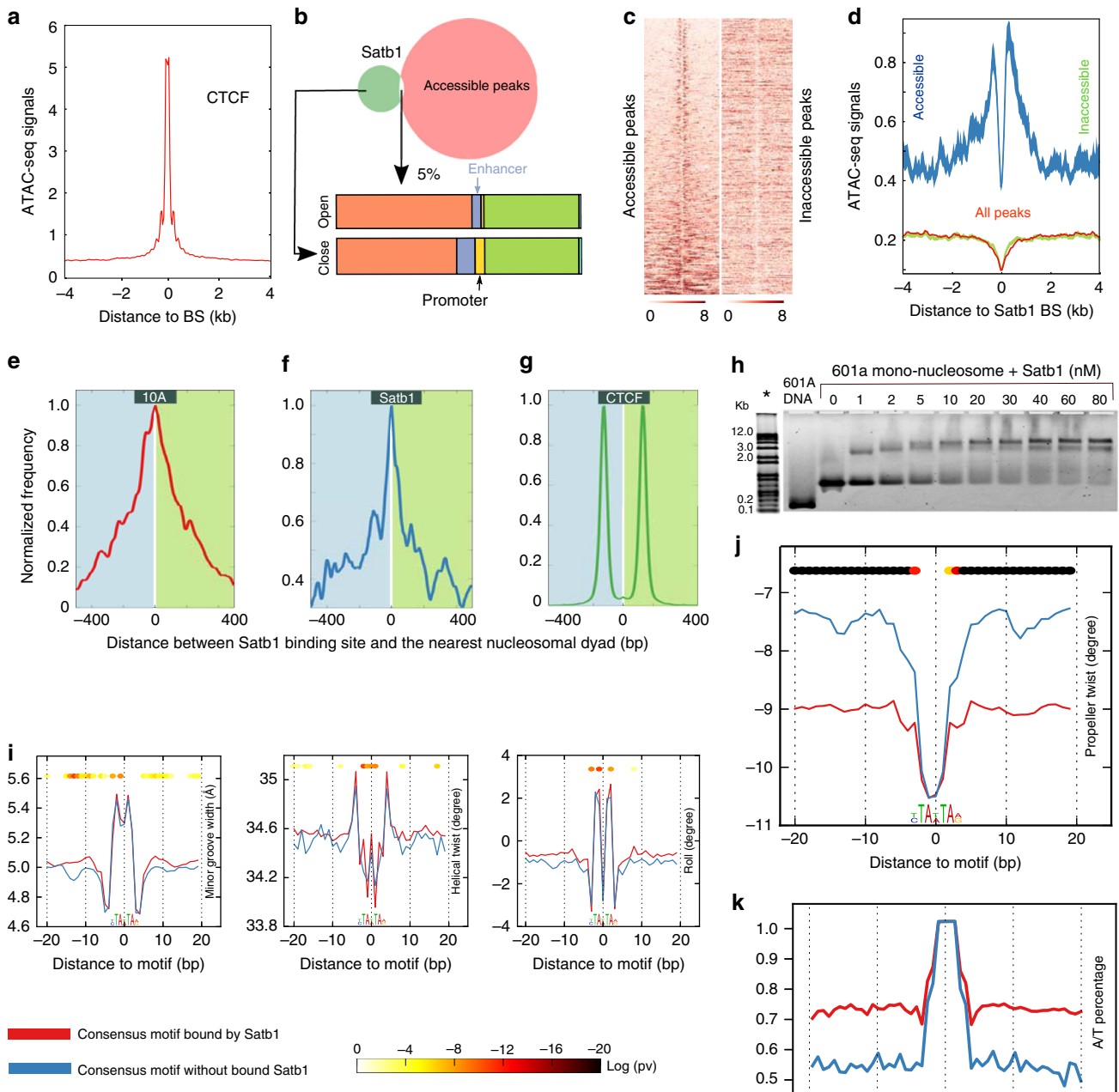

**Fig. 3** High nucleosome-density and negative propeller twist characterize Satb1 binding sites. **a** The average of ATAC-seq signal from an 8 kb window centered on CTCF binding sites in transposase accessible and inaccessible regions of the genome. **b** Genome annotation for accessible (~5% of all binding sites) and inaccessible Satb1 binding sites (~95% of all binding sites). **c** The ATAC-seq signal from an 8 kb window centered on Satb1 binding sites in transposase accessible and inaccessible regions of the genome. **d** The average of tagmentation signals at Satb1 binding sites in accessible and inaccessible regions. **e, f** Distribution of distances from nucleosomal dyad to Satb1 binding centers in native MCF-10A cells which lack Satb1 (**e**) and cells expressing exogenous Satb1 (**f**). **g** Distribution of distances from nucleosomal dyad to CTCF binding centers in MCF-10A cells. **h** 601 A nucleosome-EMSA shows that Satb1 binds efficiently to nucleosome-core embedded consensus motif resulting in 3 distinctly shifted populations, each increment presumably representing binding to one of the sites incorporated in SH3, SH4, and SH6. The ladder (* aligned) image was obtained by staining the EMSA gel with SYBR Green after the fluorescence image was acquired in the Alexa 488 channel. **i, j** Line plots showing average of MGW, HelT, Roll (**i**) and ProT (**j**) values across 40 bp windows centered on consensus motifs bound by FL Satb1 in nucleosome free regions (n = 1968) compared to consensus motifs free of Satb1 (down sampled from n = 3039). **k** Line plots showing average percent A/T content across the same 40 bp windows as **i, j**. p-values in **i, j** are obtained using Mann-Whitney U test. Note: Total number of motifs is much larger than total number of CHIP-seq peaks, since many binding sites have more than one motif. Source data are provided as a Source Data file

upon CPT treatment and to a much greater extent upon ICRF-193 treatment (Fig. 4d). Upon expression of exogenous Satb1, the stark deficiency in Psoralen-crosslinking at Satb1 binding sites seen in native MCF-10A cells treated with CPT and ICRF-193 was abrogated (Fig. 4e). This suggests that Satb1 prefers

torsionally stressed DNA and stabilizes binding sites against helix destabilization.

Since torsional stress regulates chromatin fine structure[45], we asked whether there are distinct classes of torsional states that are suited for specific chromatin-protein interactions. A genome

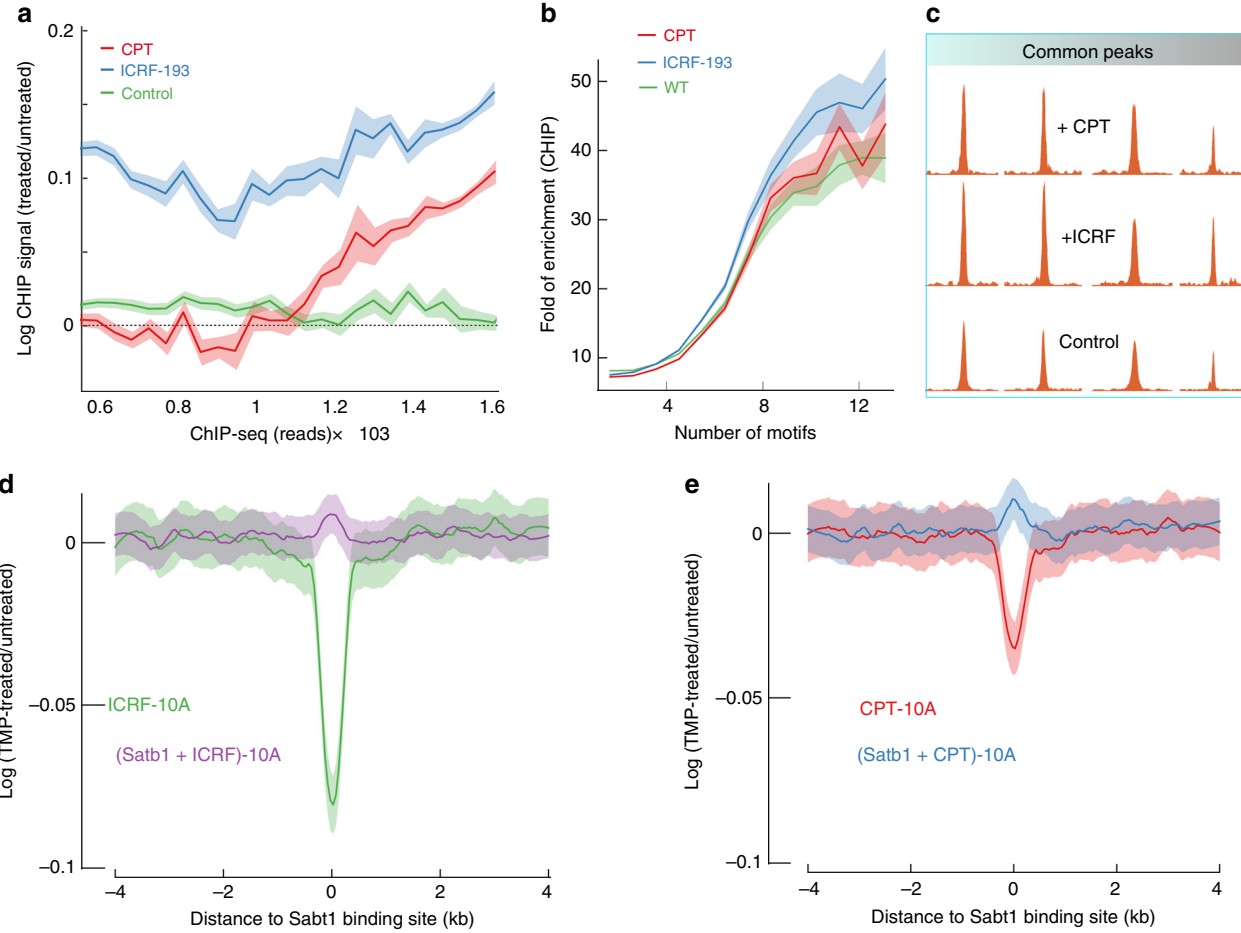

**Fig. 4** Satb1 binds better under enhanced negative torsional stress and stabilizes BURs. **a** Satb1 ChIP-seq signal ratio of "MCF-10A + Satb1" cells treated with Topo I or II inhibitors, ICRF-193 (blue) and CPT (red), respectively, over untreated "MCF-10A + Satb1" cells, plotted against increasing raw ChIP-seq reads. The green line shows the ratio between two wild-type replicates. The logarithm plot of ratio shows increased Satb1 binding upon treatment. **b** Plot of ChIP-seq signal of ICRF-193 (blue), CPT (red) treated "MCF-10A + Satb1" cells versus motif density calculated from a 400 bp window centered on Satb1 binding sites. **c** Few representative examples of ChIP-seq peak intensity before treatment (lower panel) and after treatment with ICRF-193 (middle panel) and CPT (upper panel). **d, e** TMP-seq was performed on native MCF10A cells and cells expressing exogenous Satb1 in presence and absence of CPT and ICRF-193 respectively. Final TMP-seq signal was derived after correction for free genomic DNA-sequence bias at 50 bp resolution across a 4 kb window centered on Satb1 binding sites. Ratio of TMP-seq signals in "MCF-10A + Satb1" cells treated with CPT (**d**) or ICRF-193 (**e**) over untreated "MCF-10A + Satb1" cells. All error envelopes represent standard error of mean. Source data are provided as a Source Data file

wide correlative analysis of CHIP-seq read strength versus TMP signal enrichment (fold enrichment over genome average, see Methods) of 10 different chromatin binding proteins revealed distinct patterns (Fig. 5a–b). Fos, Jun and JunB, which are known to bind nucleosome-depleted regions (known as formaldehyde-assisted isolation of regulatory elements (FAIRE))[48] encompassing promoters and enhancers, showed strong preference for under-wound DNA. BRD4[49], an epigenetic reader of histone acetylation, showed no torsional bias, whereas BRCA1[50], a DNA damage repair protein that binds to DNA mostly in a sequence-independent manner, showed minimal torsional bias. Chromatin architectural proteins (CAPs) such as CTCF, SMC1 and Cohesin-SA1[51] bound to sites characterized by a broader range of torsional states compared to canonical transcription factors but showed clear bias against highly under-wound DNA. Satb1 showed a visibly sharper choice for torsional states. Unlike the other CAPs, Satb1 demonstrated almost no binding to sites with TMP signal lower than the genome average and preferred slightly under-wound DNA, with binding strength decaying sharply with increase in TMP crosslinking.

**Domain contribution to binding affinity and specificity**. A mechanistic explanation of the context sensitivity of how Satb1 interacts with the chromatin requires an understanding of how a single protein can integrate multiple selection criteria. For example, consider an additive model where Satb1's three (puta-tive) DNA interaction domains (CUT1, CUT2, and HD) all contribute to affinity and specificity; alternatively, affinity and specificity could arise out of the modular organization of Satb1 where some domains could provide affinity and others would provide specificity. Since several HD family transcription factors have been shown to bind to sequence environments with enhanced negative propeller twist[52], it is plausible that the Satb1 HD might play a role in conferring specificity through the recognition of specific DNA shape parameters. To address these questions, we compared full-length Satb1 (FL) to three domain truncation mutants, one with the N terminal dimerization domain but no DNA interaction domains (N), one with the N terminus and CUT1 (N-C1), and one lacking only the HD (ΔHD) (Fig. 6a). We used a combination of imaging and genomic techniques to evaluate the ability of these different domain

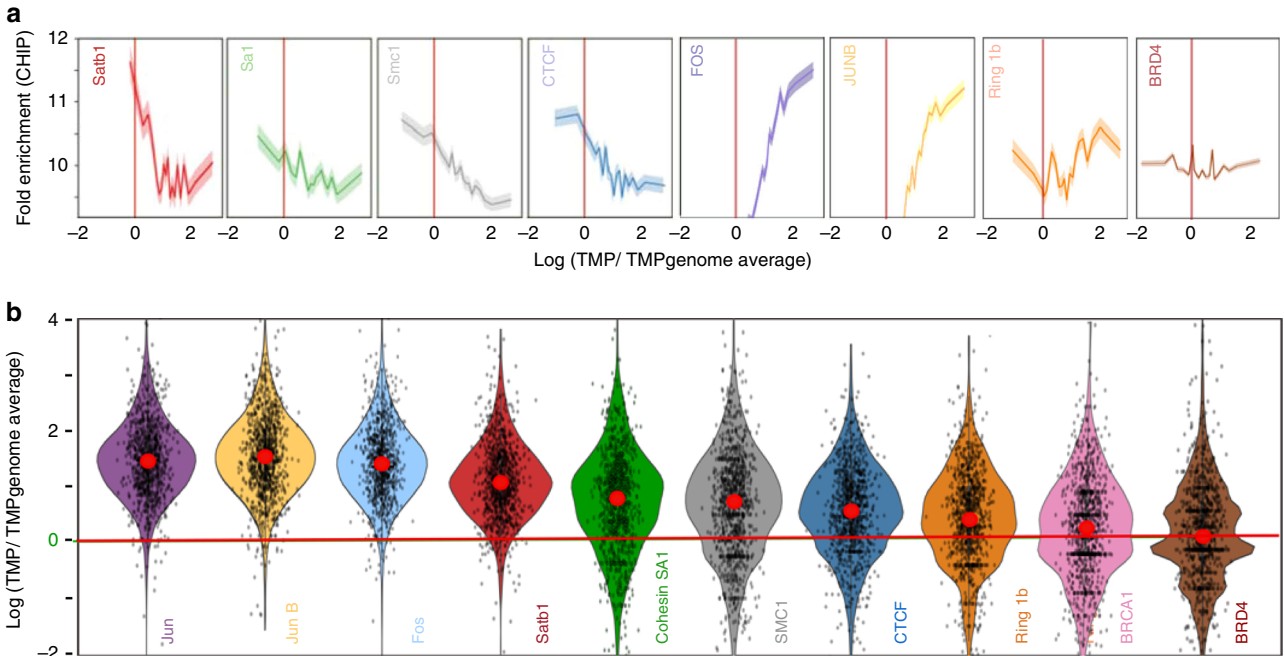

**Fig. 5** DNA torsion sensitivity of different classes of chromatin binding proteins. **a** Plots of CHIP-seq fold enrichment versus TMP signal enrichment (expressed as the logarithm of the ratio of mean TMP-seq reads of a 400 bp genomic window centered on the binding sites and the background TMP-seq signals) of 8 different chromatin binding proteins (for details see methods). All error envelopes represent standard error of mean. **b** Violin plots showing the distribution of background corrected TMP-seq signals for the top 20% strongly represented CHIP-seq peaks for 10 different chromatin binding proteins. 1000 randomly selected data points are displayed on each violin. Red dots represent the median of the distribution. Fos, Jun and JunB, are transcription factors that bind known consensus sequence motifs in nucleosome-depleted regions. BRCA1 is a DNA damage repair protein. Ring1b is a component of the Polycomb group (PcG) proteins. BRD4 binds to acetylated histones in a DNA sequence-independent manner. Satb1, CTCF, Cohesin SA1, and SMC1 are chromatin architectural proteins. Satb1 CHIP-seq data, and TMP-seq data shown here can be accessed through GEO (Accession number: GSE123292). For all the other proteins, publicly available CHIP-seq data were used. See Methods for the accession numbers. Source data are provided as a Source Data file

mutants to (a) minimally bind chromatin, (b) stably bind chromatin, and (c) specifically bind subsets of Satb1 motifs.

We generated VL3-3M2 cell lines where the native Satb1 was knocked out using CRISPR/CAS9 (Supplementary Fig. 12a, b, Fig. 6b) and was replaced with an inducible version of a full-length Satb1 or a domain truncation mutant. Spatiotemporal FRAP[32,53] was used to determine the protein's affinities to chromatin (Fig. 6c-f). The 2D spatiotemporal profiles were fit with a reaction-diffusion model[54] (Fig. 6e-f, Table 1). Notably, all constructs containing CUT1 were able to stably bind chromatin with similar dynamics, suggesting that CUT1 is the primary determinant for stable chromatin interaction. Consistent with earlier in vitro findings[40], N retained minimal binding to chromatin (Fig. 6c, e; Table 1). Spatiotemporal FRAP of different Satb1 domain constructs in MCF10A cell line showed similar chromatin binding patterns to VL3-3M2 (Supplementary Fig. 13, Table 2).

In addition to FRAP, we carried out single particle tracking of FL Satb1 and different domain truncations to directly visualize binding (Fig. 7a, Supplementary Movies 1–4). For all four cell lines, a typical cell yielded ~2000 localizations (Supplementary Fig. 14a). The comparable number of localizations across the four conditions reflects similar expression levels and uniform imaging settings. The molecular position versus time tracks (Fig. 7b) revealed two major classes of Satb1, namely slow (and likely anchored to chromatin) and fast (and presumably diffusing through the nucleus). Some trajectories showed transitions between states (Fig. 7b, 'multistate'). On a time-projection of the individual frames, the long-lasting tracks were visible as contiguous signals (Fig. 7c, d).

Since ensemble averaging obscures motion characteristics (Fig. 7e), we grouped trajectories in slow and fast fractions based on their initial displacements (Fig. 6e inset). The fast and slow MSD (mean squared displacement) data for small time intervals were well fit by lines of constant slope of ~1.02 and ~0.45, respectively (Fig. 7e inset). Single locus tracking experiments in Yeast[55] and other eukaryotes[56] have found values of ~0.5 for the exponent of locus diffusion[55], similar to our findings of ~0.45 (Fig. 7e inset). This suggests that when Satb1 is bound to chromatin, it exhibits the Rouse dynamics[55] of the slowly undulating chromatin polymer.

The effective diffusion coefficient for the slow fraction of around $0.04\,\mu m^2\,s^{-1}$ is consistent with previous measurements for chromatin fiducials[54,55,57]. The diffusion coefficient estimated for the fast fraction, around $0.15\,\mu m^2\,s^{-1}$, falls slightly below the lower bound of what has been measured for freely diffusing transcription factors (typical values of 0.5 to $5\,\mu m^2\,s^{-1}$). For each construct, we determined the distribution of dwell times (Fig. 7f–i). Consistent with FRAP data, N showed very few tracks with >1 s lifetime (Fig. 7f), suggesting that N can interact with chromatin, either directly or indirectly but does not engage stably (Fig. 7f-g; Supplementary Movie 4). The other constructs exhibited a significantly higher fraction of long-lived tracks (Fig. 7f-g, Supplementary Movies 1–3). For fast image acquisition rates, it has been shown that the slow fraction is dominated by outliers[34]. A box plot of the top 10% of longest surviving tracks showed that there are many more statistically classified outliers for FL, N-C1, and ΔHD (lasting up to 25 s) than for N (Fig. 7i). Furthermore, the survival distribution of tracks lasting greater than 0.5 s was fit better with a biexponential function[34] for N-C1

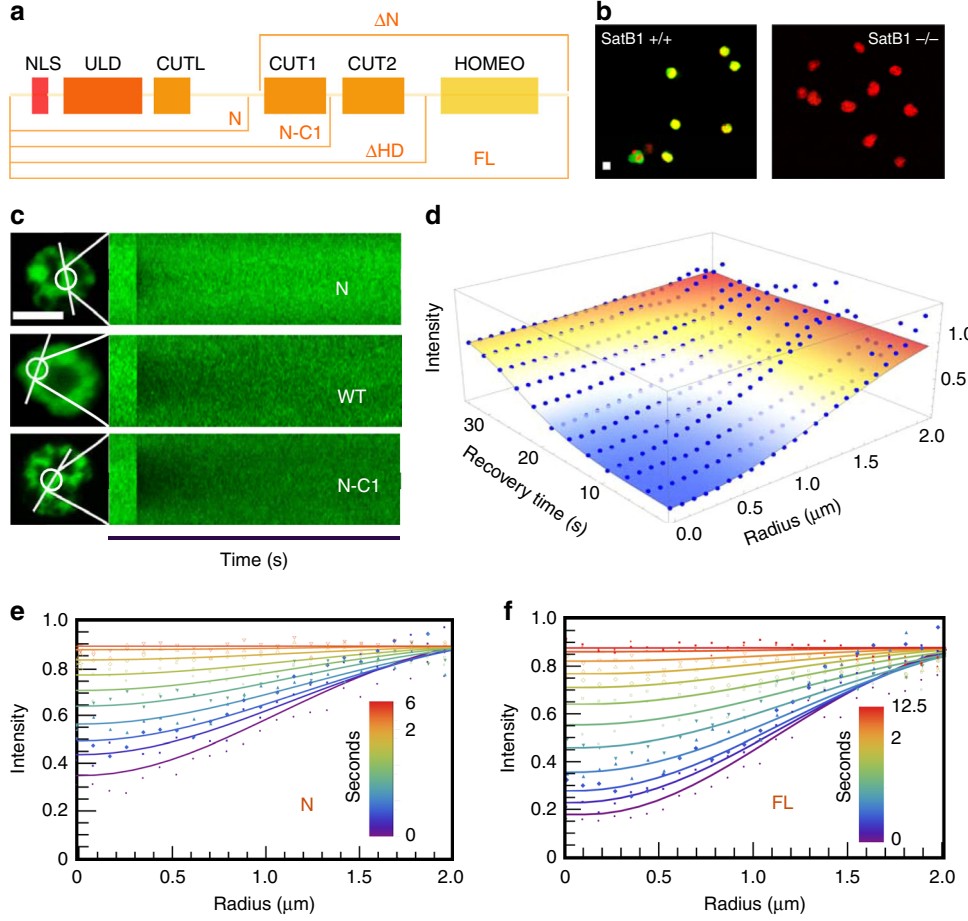

**Fig. 6** Spatiotemporal FRAP analysis of Satb1 dynamics in the nucleus. **a** A schematic representation of the domain organization of Satb1 and the various domain constructs which were generated to study Satb1 dynamics. **b** Satb1 knockout lines were generated by CRISPR/Cas9 strategy. Immunofluorescence image of WT Vl3 3M2 cells and Vl3 3M2 cells where the native Satb1 has been knocked out (Satb1 green and DNA in red) **c** Examples of time versus fluorescence intensity carpets of high resolution spatiotemporal maps of fluorescence bleaching and recovery along scan lines bisecting a bleach spot in Satb1$^{-/-}$ cells expressing exogenous eGFP fusions of FL Satb1 and different domain constructs. **d** The best-fit surface to a FL Satb1 spatiotemporal FRAP data. **e**, **f** The best fits of representative recovery profiles for N (**e**) and FL (**f**) are displayed at select times, with the color bar representing time in seconds. For estimates of spatiotemporal FRAP fit parameters, as well as total number of cells analyzed per condition, see tables 1 and 2. Scale bar is 5 μm. See source data for analysis and Supplementary Code 1 for custom analysis codes

### Table 1 Estimates of spatiotemporal FRAP fit parameters for VL3-3M2

| | Fit to a diffusion only model | Fit to a reaction diffusion model | | | | |
|---|---|---|---|---|---|---|
| | Diffusion coefficient (μm² s⁻¹) | $D_{eff}$ (fast diffusing bound fraction) (μm² s⁻¹) | Bound fraction fast | Bound fraction slow | $k_{off}$ (s⁻¹) | Number of cells |
| FL | | 2.39 +/− 1.26 | 0.50 +/− 0.15 | 0.37 +/− 0.12 | 0.38 +/− 0.26 | 39 |
| ΔHD | | 2.95 +/− 1.74 | 0.47 +/− 0.14 | 0.39 +/− 0.13 | 0.35 +/− 0.17 | 29 |
| N-C1 | | 3.12 +/− 1.15 | 0.45 +/− 0.09 | 0.39 +/− 0.08 | 0.74 +/− 0.42 | 29 |
| N | | 9.05 +/− 3.66 | 0.17 +/− 0.22 | 0.28 +/− 0.12 | 1.23 +/− 1.03 | 30 |
| ΔN | 11.53 +/− 5.86 | NA | NA | NA | NA | 32 |

### Table 2 Estimates of spatiotemporal FRAP fit parameters for MCF10A

| | $D_{eff}$ (fast diffusing bound fraction) (μm² s⁻¹) | Bound fraction fast | Bound fraction slow | $k_{off}$ (s⁻¹) | Number of cells |
|---|---|---|---|---|---|
| FL | 2.20 +/− 0.94 | 0.55 +/− 0.12 | 0.33 +/− 0.10 | 0.36 +/− 0.17 | 17 |
| ΔHD | 1.96 +/− 1.03 | 0.57 +/− 0.11 | 0.32 +/− 0.08 | 0.28 +/− 0.17 | 19 |
| N-C1 | 2.96 +/− 1.40 | 0.52 +/− 0.13 | 0.3 +/− 0.09 | 0.35 +/− 0.24 | 28 |
| N | 8.06 +/− 2.80 | 0.23 +/− 0.19 | 0.22 +/− 0.08 | 0.37 +/− 0.15 | 26 |

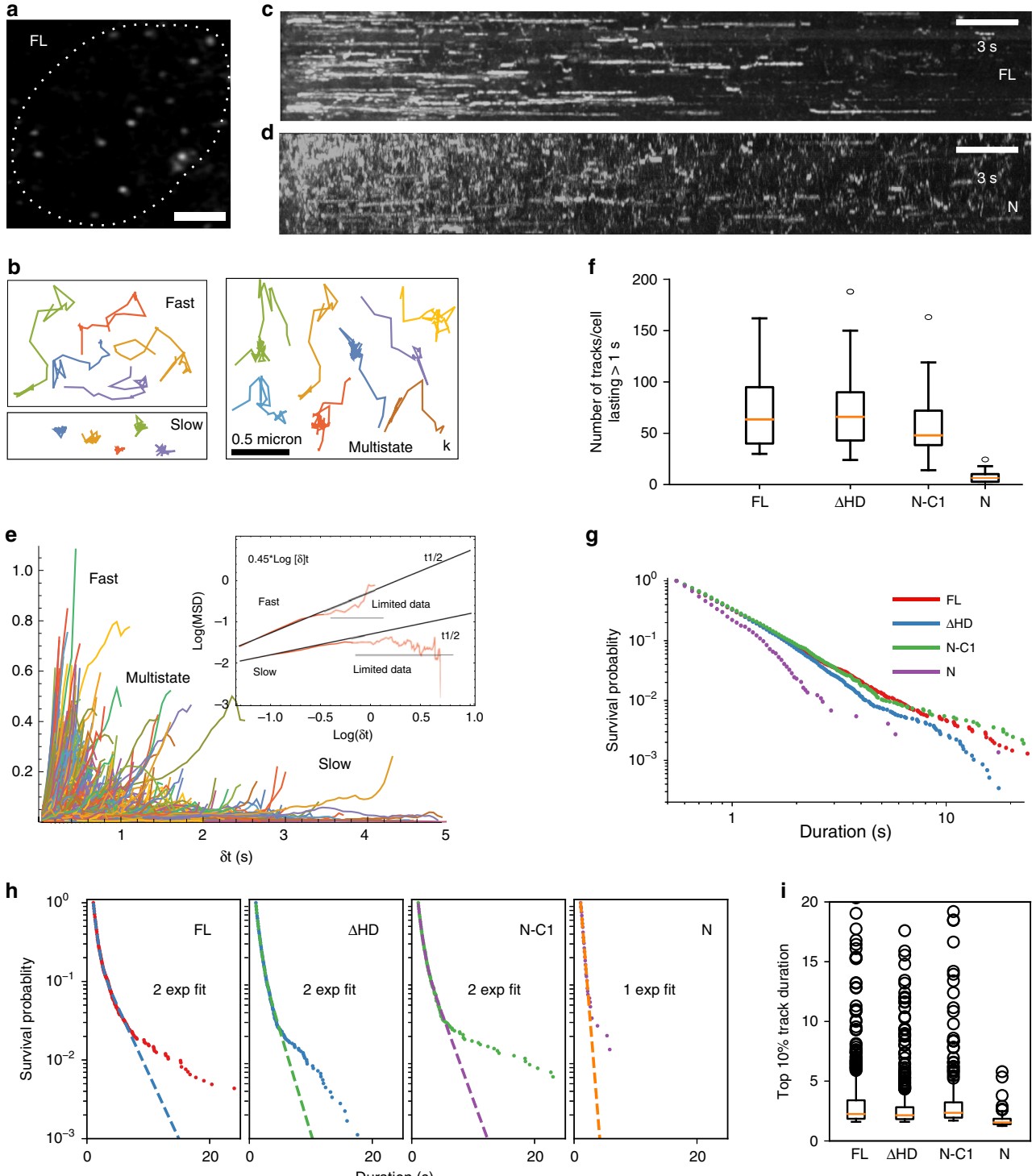

**Fig. 7** Single molecule tracking analysis of Satb1. **a** A Hilo TIRF image of an MCF-10A cell showing single Satb1-eGFP molecules (Scale bar is 5 μm).
**b** Examples of FL tracks at 20 Hz. The maximum inter-frame displacement was set to 0.25 μm which is the maximum step size for nucleosome embedded histone H2B (Supplementary Fig. 14b-d). **c, d** Time projection stacks reveal stably bound molecules as continuous stretches of signal. FL (**c**) time projection has more instances of contiguous signal than the dimerization unit N (**d**). **e** Evidence of two states and subdiffusion. **f** Long tracks are enriched with N-C1, ΔHD, and FL. Box whisker plots showing distribution of number of tracks per cell that last more than 1 s (20 consecutive frames). Median values for D (6), N-C1 (48), ΔHD (66), FL (64). Whisker represents the 1.5*IQR (interquartile range) beyond 3rd quartile (upper whisker) or below 1st quartile (lower whisker). **g** Survival plot of track durations. **h** For FL, N-C1 and ΔHD the survival distributions are fit better with two exponentials than one except for N. The rate constants and $R^2$ of the fit for each construct are as follows FL (k1: 2.08+/− 0.03 $s^{-1}$, k2: 0.39 + 0.01 $s^{-1}$, $R^2$: 0.9994), ΔHD (k1: 2.12+/− 0.06 $s^{-1}$, k2: 0.59+/− 0.03 $s^{-1}$, $R^2$: 0.9992), N-C1 (k1: 2.03+/− 0.05 $s^{-1}$; k2: 0.5+/− 0.02 $s^{-1}$, $R^2$: 0.9990), N (k: 2.22+/− 0.04 $s^{-1}$, $R^2$: 0.9964). **i** Box-whisker plot representing the distribution top 10% residence times for the different constructs. SMT analysis was done on trajectories (>5 s) obtained from 26 cells for FL (7285 trajectories), 25 cells for ΔHD (6871 trajectories), 23 cells for N-C1 (4274 trajectories) and 20 cells for N (924 trajectories), respectively. All SMT analysis have been included in the source data file

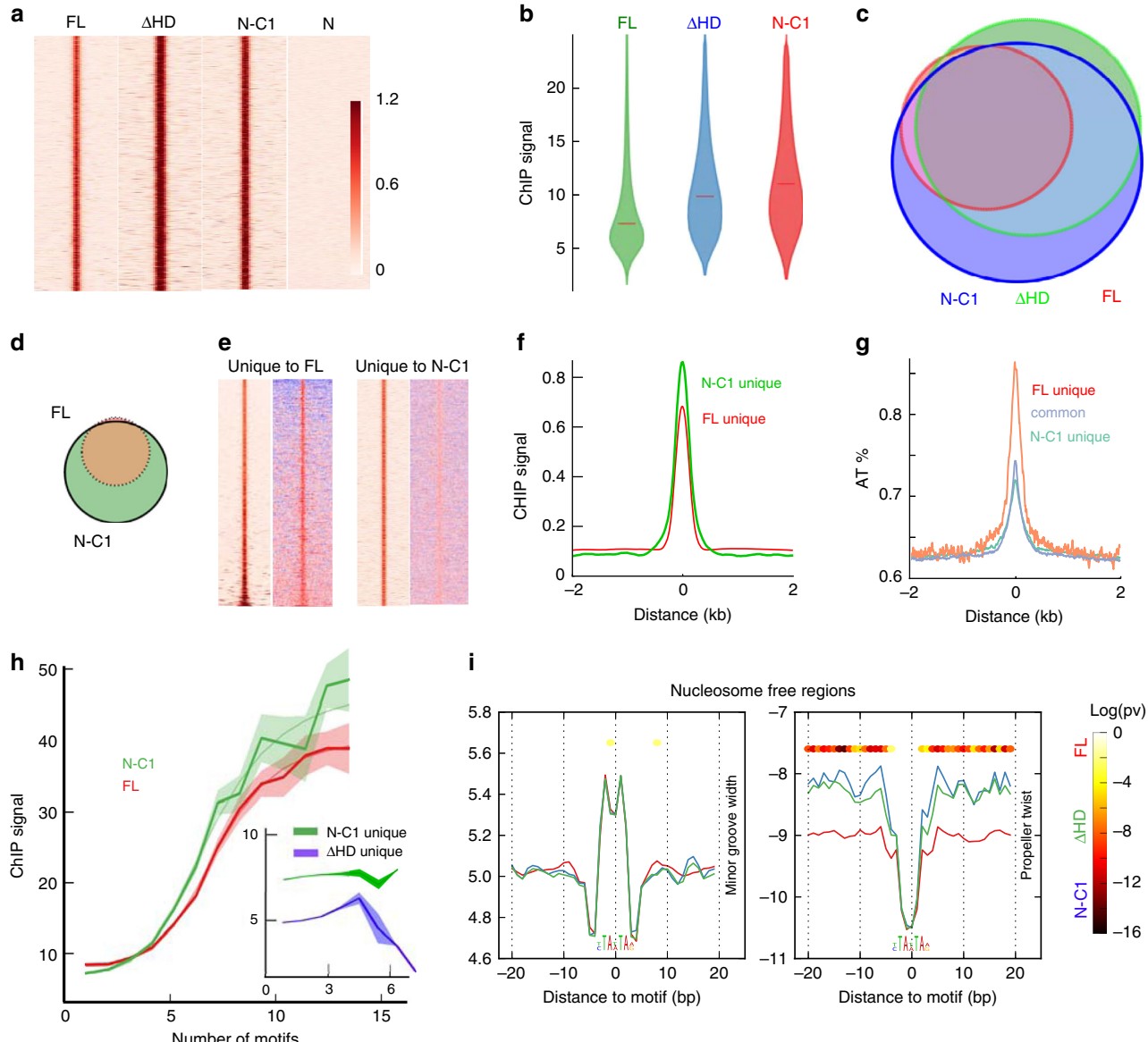

**Fig. 8** Homeo domain is dispensable for high affinity binding but increases selection stringency. **a** Heatmap of ChIP-seq signals (read pileups) along a 4 kb window centered on binding peaks of different Satb1 domain constructs. Both N-C1 and ΔHD show higher signal enrichment at binding sites than FL Satb1. The dimerization domain N on its own shows no consistent signal enrichment. **b** Violin plots showing the distribution of the peak signal strengths for 3 different Satb1 constructs (Median peak strengths are shown as horizontal lines). **c** Venn diagram showing binding site overlap between all 3 domain constructs. **d** Venn diagram of FL versus N-C1 binding sites. N-C1 binds nearly all (~97%) of all FL binding sites and more than as many sites exclusively. **e** Heatmap of ChIP-seq signals (read pileups) (left column) and AT percentage (right column) of binding sites unique to FL Satb1 and N-C1. **f, g** Average plot of ChIP-seq signals (**f**) and AT percentage (**g**) along a 4 kb window across the binding sites that are exclusively bound by FL Satb1 or N-C1. **h** N-C1 and ΔHD bind cooperatively to sites that are common with FL Satb1. However, sites exclusively bound by N-C1 and N-C1-2 show no such cooperative pattern (inset). Error envelopes are plotted using standard error of the mean. **i** Distinct DNA-shape features help define FL Satb1 and N-C1 binding. Line plots showing average of MGW and ProT values across 40 bp windows centered on consensus sites bound by FL Satb1 and sites bound exclusively by N-C1 ($n = 971$) and ΔHD ($n = 976$) in nucleosome free regions. p-values in **i** are obtained using Mann–Whitney U test. Source data are provided as a Source Data file

and ΔHD, unlike N which showed a mono-exponential distribution (Fig. 7h). Since survival plot of track durations for FL and the different domain truncations showed that the Homeo domain is dispensable for high affinity binding, we asked whether Satb1 could access target sites inside nucleosomal core sequences in the absence of a functional Homeodomain. Recombinant ΔHD showed efficient binding to 601 A mono-nucleosome in an in vitro EMSA setup (Supplementary Fig. 15).

To assess if HD imparts specificity, we carried out ChIP-seq on MCF10A cells inducibly expressing eGFP fusions of all four constructs. To ensure that the expression level of the different

domain constructs was comparable, we sorted the different cell lines for an identical fluorescence gate (Supplementary Fig. 2). The ChIP-seq data replicates were highly correlated for all the domain constructs (Supplementary Fig. 16a). In line with the single molecule data, all three constructs containing CUT1 engaged stably with chromatin (7.4 to 11.1-fold enrichment over the background, Fig. 8a-b, Supplementary Fig. 16b). The slightly more enriched occupancy of ΔHD and N-C1 compared to FL at the commonly bound sites could be due to small differences in accessibility of the GFP epitope in the different Satb1 domain constructs or due to posttranslational modifications that are

unique to HD. Alternatively, FL binding could alter the epigenetic state of the underlying chromatin even upon short exposure. The N construct pulled down comparable amounts of DNA as the other mutants but had no consistent binding pattern across the genome (Fig. 8a). Interestingly, FL bound to significantly fewer genomic sites (~22,000) than N-C1 and ΔHD (~40,000 and 47,000, respectively, Fig. 8c, d). Motif analysis using MEME[39] revealed that all three constructs bound essentially the same core sequence motif (Supplementary Fig. 16c) and that the majority of these sites were in intronic and intergenic regions of the genome (Supplementary Fig. 16d). However, the subset of sites that were exclusively bound by N-C1 and ΔHD had a lower A/T percentage than FL (Fig. 8e–g, Supplementary Fig. 17a-c), lower motif density, and showed no signs of cooperativity (Fig. 8h inset). CHIP-seq analysis of chromatin binding by the different Satb1 domain constructs in VL3-3M2 revealed essentially identical trends (Supplementary Fig. 18a-e). DNA shape calculations revealed that these sites were characterized by a more positive propeller twist angle and lesser A/T content in their immediate flanks. This was true for both nucleosome-free regions (Fig. 8i) and inaccessible regions (Supplementary Fig. 19a, b). In summary, these data indicate that CUT1 is sufficient for high-affinity binding whereas HD primarily aids in specificity through shape readout of motif flanks.

## Discussion

Our data provide the first comprehensive quantitative picture of how Satb1 uses multiple identifiers to selectively target a small cohort of potential binding sites in the genome. SMT and spatiotemporal FRAP revealed that the CUT1 domain is enough to ensure prolonged interactions with the genome. By complementing imaging with deep sequencing, we have found that while HD is dispensable for prolonged interactions in SMT experiments, it plays a crucial role in assigning specificity by selecting targets surrounded by sequences with more negative propeller twist and higher A/T percentage. Further, Satb1 shows cooperative binding to genomic regions with high binding site density, a feature that is also visible in dynamic super-resolution images as chromatin interaction hotspots. While long tracts of poly(dA:dT) are unfavorable to nucleosome formation[58], bendable di-nucleotides (AT, TA) frequently occur periodically at helix/histone interaction sites along the length of a nucleosome[58]. Interestingly, Satb1 binding sites also show ~10 bp periodic spacing and Satb1 preferentially targets closed chromatin by directly accessing nucleosome embedded motifs in vivo and in vitro. Indeed, enhancement in Satb1 binding efficiency upon treatment with CPT and ICRF-193 is consistent with a nucleosome mediated cooperativity model[59], as treatment with topoisomerase inhibitors would destabilize nucleosomes, facilitating partial unwrapping. Torsional stress sensing and preferential binding to nucleosome embedded sites may constitute the effector and response modules of Satb1's binding mechanism. It has been suggested that recognition of partial motifs on one face of nucleosomal DNA is the discerning feature of a pioneer factor[41]. Further studies are needed to delineate the Satb1-nucleosme binding interface at high resolution and to temporally resolve the changes in chromatin architecture upon binding of Satb1 to silent chromatin.

## Methods

**CRISPR**. We knocked out native Satb1 in Vl3 3M2 with CRISPR-mediated homogeneous recombination technique. Source vectors of guide-tracer DNA and Cas9 was obtained from Doudna lab at Berkeley and Zhang lab at MIT, through Addgene. Briefly, we first designed a CRISPR guide vector with double BsaI sites that is able to insert target oligo efficiently using golden gate method. This vector was constructed using SLIC[60]. The donor plasmid was built based on pmCherry-C1 vector where the DNA of two homologous arms flanking the sequence of

interest was inserted. Here mCherry was used a negative selection marker as it would be removed if proper homologous recombination and integration took place. The HR cassette consisted of a 5' end and 3' end homologous arms of 800–1000 bp (PCRed directly from Vl3 3M2 genomic DNA) flanking the loci of interest (for KO: puromycin-BGH polyA, and for HR eGFP fusion at 3' end: flexible linker-eGFP-P2A-puromycin). Surveyor assay was performed for each guide sequence according to Sanjana et al.[61]. HR donor plasmid, guide DNA plasmid and Cas9 cDNA plasmid were electropolated into Vl3-3M2 cells using Neon system (invitrogen). Double nickase version of cas9 was used to enhance the editing specificity[62]. The cells were then selected with puromycin after 3 days of culture and sorted into 96-well for clonal selection. Cells per well was immuno-stained to obtain the final clonal Vl3 3M2 cells with double-allele knock-out. To tag the native Satb1 with eGFP in Vl3 3M2 cells, we used the same technique to insert an eGFP-P2A-puromycin sequence cassette replacing the stop codon by homologous recombination.

Satb1 knock-out CRISPR sgRNA (antisense strand): AGTTGCCTCGTTCA AATGA

Satb1 eGFP knock-in CRISPR sgRNA (antisense strand): GTTCTCTCAGT CTTTCAAGT

**Plasmids and cell lines**. FL Satb1 and the different domain deletion mutants were generated as C terminal eGFP fusions in a pLenti based vector with cumate switch promoter and stably integrated into VL3 3M2 (gift from Stephen T. Smale) Satb1$^{-/-}$ or MCF-10A (ATCC® CRL-10317™) cell lines. Stable cell lines were generated by lentiviral transduction. Cells were then selected for appropriate resistance marker to achieve stable integration and sorted by FACS.

**Protein purification**. Recombinant pETDuet plasmids were transformed into SoluBL21 E. coli (Genlantis, cat. C700200). E. coli were grown in M9 minimal media (according to manufacturer's protocol) at 16C, and protein expression was induced by IPTG. Unless otherwise stated, all subsequent steps were done at 4 C. Cells were spun down and lysed prior to protein isolation by standard His Nickel-NTA (Invitrogen, cat. R90101) column chromatography. The Nickel-NTA isolated fraction was further purified using Superose 6 Increase column 10/300 GL (GE Healthcare) size-exclusion chromatography. The choice of column provided the appropriate fractionation range and resolution such that the expected protein tetramer could be purified from smaller incomplete or degraded products. Sub-sequent analysis of column fractions was performed by visual inspection of SDS-Page gel electrophoresis (Invitrogen, cat. NP0322) and fractions meeting expected size and purity were pooled and concentrated (50 mM TrisHCl, 100 mM NaCl, 1 mM EDTA, 1 mM DTT, 10% v/v Glycerol, pH = 7.3 at 25 °C).

**Nucleosome reconstitution and EMSA**. Core histone octamers were isolated from chicken erythrocytes[63,64]. All buffers used for purifying core histones were sup-plemented with 0.5 mM PMSF, 1000× final dilution of protease inhibitor cocktail (Sigma, #P8340), 20 μg/ml TPCK (Sigma, #90182), 0.5 mM Benzamidine (Sigma, #B6506). Chicken erythrocyte nuclei were first washed with RSB buffer (10 mM NaCl, 3 mM MgCl 2, 10 mM Tris-HCl, 0.5% NP-40, pH 7.5) and then resuspended in RSB supplemented with 1 mM CaCl2 and then treated with 3 μ/ml Micrococcal Nuclease for 20 min at 37 C. The nuclease digestion was stopped by adding 5 mM EDTA followed by incubation on ice. Digested chromatin was spun down at 8000 RPM for 10 min and the supernatant was discarded. The pellet was washed twice in 20 ml TE buffer (10 mM Tris-HCl, pH 7.5, 0.25 mM EDTA plus inhibitors). The final pellet was resuspended in TE buffer supplemented with 0.3 M NaCl and centrifuged at 3500 rpm for 10 min The pellet was resuspended in TE plus 0.65 M NaCl, homogenized in Dounce Homogenizer and then spun for 16 h at 46,000 rpm over a sucrose cushion (15% sucrose in TE plus 0.65 M NaCl). H5 histones were removed by repeating the wash in 0.675 M NaCl. The pellet was homogenized in TE buffer containing 2.0 M NaCl and centrifuged again for 16 h at 46,000 rpm. The supernatant contained equimolar mixture of 4 core histones.

Nucleosomes were assembled using 5' Alexa 488 labeled 601 A which is derivative of the nucleosome positioning 601[43]. The nucleotides at superhelical locations SHL3, SHL4, and SHL6 (where the minor groove is facing histones/histone core) were modified (shown below in bold) to generate Satb1 consensus sites.

5'CTATACGCGGCCGCCCTGGAGAATCCCGGTCTGCAGGCCGCTCAA TTGGTCGTAGACAGCTCTAGCACCGCTTAAACGCACGTACGCGCTGTCC CCCGCGTTTTAACCGCCAAGGT**TAATA**ATCCT**TAATA**ACCAGGCACGT GTCAT**TAATA**AACATCCTGTGCATGTGGATCCGCACTC3'

Reconstitution was performed following established protocols[64].

Briefly, core histones and DNA template were mixed at equimolar ratios in low binding tubes (Fisher Scientific, cat. #02–681–311) in TE buffer (plus 2 M NaCl) and incubated on ice for 1 h. The mixture was then injected into a Slyde-A-Lyzer unit (3.5 K MW cut off, Thermo Scientific, cat. #69552) and dialyzed in a reconstitution buffer (20 mM Tris.HCl pH 8.0, 0.5 mM EDTA, 0.1% NP-40, 1 mM DTT, 0.5 mM Benzamidine) with a stepwise drop in NaCl concentration (1.0 M NaCl for 3 h, 0.75 M NaCl for 3 h, 0.5 M NaCl for 12 h, 50 mM NaCl for 6 h) to generate final reconstitutes.

For Electrophoretic Mobility Shift Assays 6 nM nucleosomes were incubated with increasing loading of Satb1in total volume of 25 μl in STE buffer (80 mM NaCl, 20 mM Tris-HCl, pH 8.0, 0.01% NP-40, 1 mM DTT). Resulting complexes were separated on 1% type IV (Sigma cat. #A3643) agarose gel electrophoresis in 0.5xTBE buffer at constant 100 V and visualized on Typhoon fluoroimager.

Uncropped versions of the EMSA gels shown in Fig. 3h and Supplementary Fig. 15 have been included in the source data file.

**Image correlation spectroscopy.** Image correlation spectroscopy was conducted according to Cardarelli et al. 2010[33]. Zeiss LSM 700 laser scanning microscope was used for image acquisition using ×63 oil immersion NA 1.40 objective. A 3.2 μm-line was scanned repeatedly with a pixel dwell time of 6.3 μs (32 pixels) and a line time of 0.47 ms. Data were processed by the SimFCS software developed at the Laboratory for Fluorescence Dynamics.

**Single molecule imaging.** All live cell single molecule imaging was done using HiLo Total internal reflection microscopy (HiLo-TIRFM)[38], on an Olympus CellTIRF system. TIRF angles for each fiber-coupled illumination laser were controlled independently with a motor. Imaging was done through a 1.49 NA ×100 objective and ×1.6 Optovar magnifier and recorded on an Andor iXon Plus EMCCD camera at 20 Hz. All dichroics and filters were purchased from Semrock. All experiments were performed within a heated, CO2 controlled incubation chamber set to 37 °C and 5%, respectively with additional temperature control provided using a collar-type objective heater, also set to 37 °C.

**Single Molecule tracking and analysis.** Single molecule tracking was preformed using Mathematica and python. Detectable particles in each frame were first identified by performing a Laplacian of Gaussian filter and then thresholding the filtered image based on intensity. Then the raw image data of each identified particle was fitted with a 2D Gaussian function to obtain particle $x/y$ position and signal amplitude. The single particle positions in each frame were then linked into single particle trajectories based on nearest-neighbor algorithm with a defined maximum per-frame jump distance. The maximum frame to frame displacement was set to 0.25 μm as this was the maximum step size that we observed for the chromatin fiducial histone H2B. Finally, a quality control step was performed where a large subset of particles and trajectories were validated by eye. Trajectories that lasted ≥ 10 frames (0.5 s) were selected for further analysis. Individual Satb1 molecules showed considerable variability in their motion. We classified displacements into fast, slow, and multistate categories based on the initial MSD values of each trace. The first 10 Log[δt] points (corresponding to δt = 500 ms) were fit to a line (Log[MSD] = a + b*Log[δt]).

For super-resolution imaging, after localizing the fluorophores, we combined all the localization centroids for the full length Satb1 in one computed image. For each detected molecule, a normalized symmetric 2D Gaussian (with a standard deviation equal to the computed localization uncertainty, which is on order of 30 nm) was drawn, and all the Gaussians from all the localizations (collected over 35 s at 20 Hz) were summed to yield a super-resolution image.

**Spatiotemporal FRAP imaging and analysis.** For FRAP experiments VL3 3M2 cells expressing different domain constructs were grown in RPMI media and then embedded in CyGEL Sustain (#CS20500) Biostatus for live imaging. FRAP experiments were performed on a Zeiss LSM 700 confocal microscope with a ×63/1.40 oil immersion objective. Cells were equilibrated at 37 °C with 5% CO2 for 20 min before imaging. For different cell lines, we scanned a 50 × 2-pixel rectangle (pixel size = 0.0893 μm) inside the thymocyte nucleus in euchromatic regions with homogeneous fluorescence distribution using 488 nm laser with 0.6% power and PMT gain 825. The cells were imaged at a frequency of 0.0076 s/frame. A circular spot with radius = 15 pix (1.34 μm) was bleached so that the center of the circle aligned with the center of the scanned rectangle. Experiments were repeated in the same cell nucleus one more time without the bleaching step to correct for photo-bleaching[54].

The FRAP images were corrected for photo-bleaching and then normalized. Spatiotemporal fluorescence data for fitting were generated for each time point by averaging over the two pixels that constituted the height of the rectangular scan region. This resulted in a 50 × 1 pixel-wide line (VL3-3M2) or a 80 × 1 or alternatively a 160 × 1 pixel-wide line (MCF10A) that represented the fluorescence along a line bisecting the bleach spot. This line was then averaged about the bleach center to produce a line that represented the radial distribution of the bleach spot fluorescence. These data were exponentially sampled in time and fit with a reaction diffusion model[32,54]. Three parameters: $D_{eff}$, $k_{on^*}$ and $k_{off}$ were extracted from the model. Data processing and model fitting were designed and implemented in *Mathematica*.

The FRAP parameters extracted from the fit have been elaborated in Stasevich et al.[32]. Briefly, FRAP recoveries of Satb1 variants were first fit to a pure diffusion model, which yielded the diffusion constant $D$. Of the five Satb1 truncation mutants, only ΔN was well fit by the diffusion model. The other four Satb1 mutants yielded diffusion coefficients that were too slow to be explained by pure diffusion. Therefore, their FRAP recoveries were fit with a reaction-diffusion model and yielded estimates of $D_{eff}$ (effective diffusion constant). The remaining parameters $b_f$

(bound fraction in the fast diffusing state), $b_s$ (bound fraction in the slow diffusing state), $k_{on^*}$ (association rate), $k_{off}$ (dissociation rate), and $K_d$ (dissociation constant) reported in Table 1 were calculated from the fit parameters as explained in Stasevich et al.[32].

**Genomic library preparation.** ChIP-seq libraries were prepared according to Myers lab protocol[65] with modifications to adapt to chromatin shearing using Covaris AFA. Briefly, the cells were fixed with 1% PFA for 5 min, washed twice with PBS and then quenched with glycine (125 mM) for 5 min Cells are lysed with cell lysis buffer (0.5% Triton-x100, 0.25% NP-40%, 50 mM HEPES, 150 mM NaCl, 1 mM EDTA, pH 7.5) for 15 min, and sheared in TE 10 mM + 0.1% SDS buffer with Covaris (Covaris Inc) S2 according to manufactures protocol (optimal condition for fragmentation in 1 ml vial was obtained at power intensity level 4 for 10 min).

After centrifuging the lysates to remove debris, 2–5 μg antibody (Abcam #ab290, #EPR3895) was added to 4 ml of cell lysate and incubated overnight at 4 °C. Protein G magnetic beads (Invitrogen) were then added and the mixture was shaken for 2 h before collecting using a magnetic rack. The samples were washed 6 times (thrice with low salt buffer, once with high salt buffer, once with LiCl buffer and once with TE buffer) according to Myers lab protocol. The library was generated with Ultra II DNA kit (NEB) Briefly, DNA fragment ends were generated enzymatically and adaptors were ligated to immunoprecipitated DNA fragments for downstream PCR amplification using Illumina index primers.

TMP-seq libraries were prepared according to Teves and Henikoff (2014)[13] with some modifications: Cells were incubated with Trimethylpsoralen (TMP, Sigma-Aldrich T6137) at a final concentration of 2 μg/mL for 10 min in the dark. Plates of cells were then exposed to 3 kJ/m2 of 365-nm light (Fotodyne UV Transilluminator 3–3000 with 15-W bulbs). Cells were washed with PBS 1×, scraped from the culture dish and then spun down. The pellet was then resuspended (buffer containing PBS 1×, 0.2% TritonX100 and RNaseA) and incubated at 37 °C for 30 min Cells were then spun down and resuspended in TE (Tris 10 mM, 1 mM EDTA) with 0.5% SDS and proteinase K and incubated at 50 °C for 1 h. Chromatin was then sheared with Covaris to get 200–500 bp fragments and subsequently cleaned with columns (Fisher GeneJET kit).

Cross-linked fragments were enriched by repeated rounds of denaturation and Exonuclease I (Exo I) digestion following protocol from Teves and Henikoff (2014)[13]. In summary three μg of sonicated DNA, diluted to 250 μL, was boiled in a water bath for 10 min and incubated in ice water for 2 min To each sample, 30 μL of 10× Exo I buffer and 10 μL of Exo I were added, and digestion was allowed to proceed for 2 h at 37 °C. Then samples were boiled and cooled as before, and 10 μL of Exo I was added for a second round of 1-h digestion. After cleanup, Exo I–digested DNA samples were subjected to enzymatic reactions for end polishing and 'A' tailing. Illumina adaptors were ligated using NEBNext Ultra library prep kit. After that, the 5′ strand was digested with 20 U of λ exonuclease (NEB) for 30 min at 37 °C. The DNA was purified with Ampure beads. The resulting 3′ strand was used as a template for ten rounds of primer extension in 1× pfu Ultra II HS buffer, 0.8 mM dNTP, 1U of pfu UltraII HS DNA polymerase (Agilent) and 40 nM of P7 extension primer. The resulting single-stranded products were purified with Ampure beads and eluted.

The purified products were then appended with ribo-G in 1× terminal deoxynucleotidyl transferase (TdT) buffer, 10 U TdT (NEB) and 0.1 mM of rGTP in 37 °C for 30 min The products were purified with Ampure beads. The single-stranded ribotailed products were ligated to a double-stranded adaptor with CC C-overhangs (CCC overhang oligo 1: ACACTCTTTCCCTACACGACGCTCTTC CGATCTCCC, CCC overhang oligo 2: (Phosphate)AGATCGGAAGAGCGT CGTGTAGGGAAAGAGTGT) and final library was generated by PCR amplification (using NEB Q5 High-Fidelity 2× master mix) for 5–10 cycles.

ATAC-seq.: 50k cells were collected and washed with ice cold PBS and then further washed with RSB buffer (10 mM Tris, 1 mM NaCl, 3 mM MgCl2, pH 7.4) with 0.1% NP40, and tagmentated with 2.5 μl Tn5 enzyme. The libraries were prepared as per Buenrostro et al. (2013)[37]. Briefly DNA was cleaned with MiniElute columns (Qiagen), and PCR amplified for 8–12 cycles using Illumina NextEra primers (Illumina), where the cycle number was determined by qPCR.

**ChIP-ATAC seq.** We added a ChIP step before preparing for ATAC-seq sample to enrich the reads around Satb1 binding sites for determining nucleosome positions nearby. The Satb1 ChIP was performed as described above except the Covaris shearing time was reduced to 8 min After bead assisted pull down, the sample was washed once with low salt buffer, and once with RSB buffer, and resuspended in tagmentation buffer with 2 μl of Tn5 enzyme mix. The sample was then incubated at 37 °C for 15 min and cleaned with DNA clean up kit (Zymo Research). Final library was prepared the same way as ATAC-seq library.

**Genomic data analysis.** Adapter sequences in sequencing reads were trimmed with cutadapt. Quality of reads were assessed by fastqc and aligned to human genome version hg19 and mouse version mm10 with bowtie2. Typically, alignment rate was between 90–95%.

ChIP-seq binding site peak calls were made with MACS2[66], using Poisson statistical model. Differential binding was analyzed with DESeq2 package in R[67],

using binomial statistic model. The widely used MEME suite tools were employed for motif analysis[39]. Specifically, de novo motif discovery was performed with MEME and FIMO was used to search for occurrence of known motifs with default threshold.

Briefly, Motif analysis was performed using FIMO tools and custom python script. A 400 bp window centered at each binding site was selected and fasta sequence was retrieved with bedtools. FIMO was used to search for motif occurrence and location. Then the spacing between nearest motifs for each binding site and number of motif occurrences per 400 bp was calculated to correlate with ChIP-seq signals.

Other ChIP-seq data were collected from ENCODE project, with the following access numbers:ENCFF000WJE (RelA), ENCFF002CIA (TCF3), ENCFF002CJF (Oct4), ENCFF002CHF (NFATc1), ENCFF002CJA (nanog), ENCFF231TGQ (IRF2), ENCFF388AJH (IRF1), ENCFF856FVT (FoxA1), ENCFF002DCM (CTCF). The PWM (power weight matrix) for other transcription factors and chromatin regulators were retrieved from MEME motif database.

Homer software[68] was used for genome feature annotation, and generation of read heatmap. ChIP-seq signals were presented as either raw reads or fold enrichment over background derived from MACS2 peak calling, as noted in the text and figures. When using raw reads, signal strength was calculated by summing the ChIP-seq reads within a defined genomic window as noted in the main text and normalized against a total count of 10 M reads.

For TMP-seq analysis[13] the first 4 bases were trimmed in each read before alignment to remove the CCC of ligated adapter. After alignment, cross-linked sites (starting base of each sequencing read) were detected and expanded on either side by 20 bps to smooth the distribution and converted to bigwig files for calculation. A window of 50 bp was used to average the TMP-seq signals.

TMP-seq reads from purified MCF10A genomic DNA were used to correct for any sequence bias in psoralen crosslinking efficiency. Briefly, the TMP-seq reads for each experiment were first normalized to the total read count (10 M reads); then those reads were divided by background reads over a 50 bp sliding genomic window to calculate the true background corrected TMP-seq signals.

For correlative analysis of TMP-signal enrichment versus CHIP-seq strength, MCF10A CHIP-seq data for various chromatin binding proteins including Ring1B (GSE107176), cohesin complex (GSE101921), Jun, JunB and Fos (GSE115597), BRD4 (GSE72931), and BRCA1 (GSE40591) were downloaded from SRA deposits. The fastq files were trimmed of adaptors and low quality reads (Q-score < 25), and then aligned to human genome hg19 using bowtie. We use the fold of enrichment score from the peak-calling algorithm MACS2 as the TF's binding strength. The corresponding TMP-seq signal per binding site was calculated as the Logarithm of mean TMP-seq reads over a 400 bp genomic window centered on the binding sites, divided by the background TMP-seq signal (TMP-seq signal derived using purified genomic DNA as described above). The line plot for CHIP-seq versus TMP-seq relations for each TF was generated by calculating the average CHIP-seq signals for 20 evenly distributed groups of TMP-seq signal. The violin plots were used to show the distribution of TMP-seq signals for the top 20% strongly represented CHIP-seq peaks.

ATAC-seq data were analyzed according to Buenrostro et al.[37] and Denny et al.[69]. In summary, the reads were trimmed off adapter sequences by Cutadapt. Reads that were shorter than 28 bp after trimming were discarded. Remaining reads were aligned to either hg19 (human) or mm10 (mouse) genome to generate BAM files. For every library, we calculated the genome wide ratio of reads at TSS sites and reads at regions 2 kb away from TSS as the TSS-score. ATAC-seq libraries with TSS-score < 7 were discarded. MACS2 was used to call the peaks (using parameter –nomodel –keep-dup --broad). Nucleosome positions at accessible regions were called with nucleoatac tools42 with default parameters.

**DNA shape analysis**. The PWM (position weight matrix) for Satb1 motif was constructed from the motif discovery as described above. Genome wide Satb1 motif locations were obtained using Homer package (specifically the perl script scan-MotifGenomeWide.pl). The DNA sequences flanking 300 bp on either side of motif were retrieved and saved in a fasta file for DNA shape analysis using the web server[44]. All the sequences of interest and their shape parameters were categorized into 3 groups for comparison: nucleosome free region (NFR) derived from nucleosome position analysis as described above, accessible chromatin and inaccessible chromatin. When there were more than 10,000 motifs per group, motifs were down sampled randomly to 5000–7000 motifs for further analysis.

When comparing the shape parameters for weak and strong binding motifs, the shape data were mapped to ChIP-seq results to obtain the corresponding ChIP-seq signals per binding site. Typically, unless otherwise mentioned, we took 600 top and bottom motifs ranked according to ChIP-seq signals to compare the shape parameters. A nonparametric Mann–Whitney $U$ test[5] was used to assign $p$-value per DNA base with typical sample size from 300–7000 sequence elements depending on group size.

**Reporting summary**. Further information on research design is available in the Nature Research Reporting Summary linked to this article.

## Data availability
The data that support the findings of this study are available from the corresponding author upon reasonable request. Most of the data presented in Main Text Figures, as well as Supplementary Figures have been provided either as Source data files or have been uploaded to GEO under accession code GSE123292. The source data also includes a document that can be used as a roadmap to connect the source data to the corresponding figures. These source data cover Fig. 1f-h, Figs. 2, 3, 4, 5, 6, 7, 8 and Supplementary Figs. 13, 4, 5, 6, 7, 8, 9, 10, 11, 13, 15, 16, 17, 18, 19.

## Code availability
Most of the custom codes used in this article have been provided as a source code file (Supplementary Code 1).

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

## Acknowledgements

This work was partially supported by the National Institutes of Health (NIH), National Institute of General Medical Sciences (NIGMS)/National Cancer Institute (NCI) Grant GM77856, NCI Physical Sciences Oncology Center Grant U54CA143836, National Institute of Biomedical Imaging and Bioengineering (NIBIB)/4D Nucleome Roadmap Initiative 1U01EB021237 and also by NIH grant P50-HG-007735 (H.Y.C., W.J.G.). H.Y.C. is an investigator of the Howard Hughes Medical Institute. We are grateful to Davood Norouzi for suggestions regarding DNA sequence analysis.

## Author contributions

R.P.G. and J.T.L. conceived the project. R.P.G designed research. R.P.G. and Q.S. performed most experiments. M.P.R. purified Satb1 protein. R.P.G. and L.Y. performed FRAP experiments. R.P.G. designed and generated the fluorescently labeled 601 A DNA.

T.N. performed in vitro nucleosome binding assays with inputs from V.B.Z. T.J.S. provided guidance for performing spatiotemporal FRAP analysis. H.Y.C. and W.J.G. provided guidance for genomic data analysis. P.F. provided guidance for in vitro binding assays. Q.S., R.P.G., L.Y., and J.T.L. analyzed data. R.P.G. and J.T.L. wrote the paper. All authors helped with editing the paper.

## Additional information

**Competing interests:** The authors declare no competing interests.

