## [Peer Review File · Nature Communications]

Reviewers' comments:

Reviewer #1 (Remarks to the Author):

The authors found that Satb1 preferentially targets nucleosome-dense regions and binds to its consensus motif TAATA within nucleosomes. They identified genomic regions containing multiple evenly spaced Satb1 binding motifs. Additionally, utilizing Satb1 deletion constructs they identified that the homeodomain of Satb1 is dispensable for high affinity binding but is essential for conferring specificity. Most interestingly, they concluded that increasing negative torsional stress in DNA enhances Satb1 binding.

Ghosh et al have employed a multitude of techniques to fine tune their understanding of Satb1-DNA binding mechanism. These include – EMSA, image correlation spectroscopy, single molecule imaging, tracking and analysis, spatiotemporal imaging and analysis, TMP-seq and Chip-seq analysis and DNA shape analysis. Their data are very persuasive, and the authors have successfully demonstrated that Satb1 is an interesting pioneer TF that binds to the consensus sequence – TATTAGTAATAC both in the nucleosome-free as well as nucleosome occupied regions of the genome. This manuscript can be accepted after the authors have addressed the following concerns:

- In the paragraph spanning lines 141 through 158, Figure 2d is referred to prior to 2c, would a rearrangement of figure be in order?
- In the same paragraph Supplementary fig 3d is not referred to at all or anywhere else in the manuscript for that matter!
- Line 217 should read; Recombinant Satb1 robustly bound...
- Figure 3h shows the EMSA for in vitro assembled mononucleosomes and Satb1; these types of samples should be run on a 5% native PAGE which yields better resolution instead of an agarose gel as the authors have done. The native PAGE might reveal the 3 binding events on the three consensus Satb1 sequences better than the agarose gel. Also, the gel wells should be included in the image. Additionally, a DNA ladder as well as a free DNA (the same one that was used to reconstitute the nucleosomes) controls should be included in such gels
- The font size for Figure 4 parts (a-e) seems to be smaller than the rest of the figures – needs to be consistent with journal requirements
- The authors have studied the domain determinants of Satb1 binding affinity and specificity, but have not confirmed these results with their readily available in vitro nucleosomes? Did they test binding of the truncated Satb1 versions to the in vitro assembled mononucleosomes?
- Line 301 has a typo- fraction has an "r" missing
- In figure 6h, were there any R square values for the various exponential fits for survival distributions?
- In line 322, I think the authors meant to refer to Figure 7e-g NOT Figure 6e-g, also in line 323, it should be Figure 7h, inset; as well as in line 326
- In Table 1, it would be easier for readers if the authors explained BF fast and slow in footnotes, so the reader doesn't need to go back into the text to recall what those stand for. Also, it would be helpful if the authors mentioned in a separate column how many cells were analyzed for each construct
- In the movies, it would be nice to indicate the slow and fast fractions (at least an example of each) using different colored arrows or asterisks if possible

Reviewer #2 (Remarks to the Author):

In this study, Ghosh et al. examine the dynamics and binding of the transcription factor Satb1 by live cell imaging and genomic analysis. The authors tagged endogenous Satb1 with eGFP by CRISPR/Cas9 gene editing in the T cell line VL3-3M2, and they detected a highly dynamic distribution of the Satb1-eGFP fusion protein by image correlation spectroscopy. In addition, the authors expressed Satb1-eGFP ectopically in the breast epithelial cell line MCF10A by using an inducible cumate switch promoter. By ChIP-seq analysis, the authors identified ~4000 Satb1-bound sites in the T cell line and ~22000 sites in the MCF10A cell line. The authors performed ATAC-seq profiling of MCF10A and Satb1-expressing MCF10A cells, and they found that the vast

majority of Satb1 peaks are associated with transposase-inaccessible regions of chromatin. In addition, the authors examined the effects of DNA topology on Satb1 binding by treating MCF10A cells with topoisomerase inhibitors. Finally, the authors examined the effects of various truncations of Satb1 on chromatin interaction by spatiotemporal FRAP and single molecule tracking in VL3-3M2 cells, in which the endogenous Satb1 was knocked out and replaced with inducible versions of Satb1 and Satb1 domain truncations, as well as by ChIP-seq in MCF10A cells that express Satb1 or Satb1 domain truncations.

This study provides interesting and novel insights into the targeting properties of Satb1 protein by combining various imaging and genomic approaches. In general, the experiments are well performed and the data support the conclusions. However, the study would benefit from a more coherent analysis in the context of the T cell line and the epithelial cell line. For example, the authors use spatiotemporal FRAP and single molecule tracking of domain mutants in VL3-3M2 cells but perform ChIP-seq of domain mutants in MCF10A cells.

Specific comments:

- 1) The authors identify ~4000 Satb1-bound sites in T cells and 22000 sites in MCF10A cell in which Satb1 was ectopically expressed. The authors do not show whether the Satb1-bound sites in T cells are a subset of the Satb1-bound sites in Satb1-expressing MCF10A cells. The authors should provide an analysis of Satb1 protein levels in these cells.
- 2) It would be of interest to compare the Satb1 binding data with an analysis of gene expression in MCF10A cells without and with Satb1 expression and in T cells in which the endogenous Satb1 was knocked out and replaced by domain truncations.
- 3) In Figure 3, the authors show that ~95% of the Satb1 peaks in Satb1-expressing MCF10A cells are associated with transposase-inaccessible regions. Are these sites also inaccessible in MCF10A cells? This could be visualized as a heat map, similar to Figure 3C. What percentage of Satb1-bound sites in T cells are associated with transposase-inaccessible regions?
- 4) Figure 3i and 3j represent the DNA shape properties in the Satb1-bound nucleosome-free regions. This set of Satb1-bound sites represents less than 5% of Satb1-bound sites, whereas the vast majority of Satb1-bound sites is associated with transposase-inaccessible regions. Therefore, the authors should consider Figure S6 for the interpretation of the DNA shape instead of Figure 3i-k. The relative difference of propeller twist values between Satb1-bound and Satb1-unbound sites are comparable for the transposase-accessible and -inaccessible regions. How do the authors explain this observation?
- 5) The analysis of Satb1 binding under enhanced torsional stress by the use of topoisomerase inhibitors is interesting. However, the authors should consider to include other DNA-binding proteins, such as CTCF, in this analysis.
- 6) Propeller twist values are relatively low for A/T-rich dinucleotides. Does it simply reflect high A/T percentage of the Satb1-bound sites (Figures S6c and 3k)?
- 7) How many of the ~4000 T cell-associated Satb1-bound sites fall within the consensus motif not bound by Satb1 in MCF10A cells?
- 8) Figures 4a-b describe global changes on Satb1 binding on the peaks with the different read coverage. It may be helpful to extend this analysis by visualizing individual peaks as a scatterplot.
- 9) N-C1- and Δ H-D-Satb1 gain additional ~20000 occupied sites relative to full length Satb1. How do the authors explain stronger N-C1 and Δ H-D binding even on the peaks that overlap with full-length Satb1 peaks (Figure 7a)? The authors should examine protein expression of Satb1 and the domain truncations.

We thank both reviewers for their positive comments ("*Their data are very persuasive*", "*the authors have successfully demonstrated that Satb1 is an interesting pioneer TF*", "*provides interesting and novel insights into the targeting properties of Satb1 protein by combining various imaging and genomic approaches*"). We also thank the reviewers for their comments and suggestions which has helped us address key points with greater clarity. The revised draft has 1 new figure, 7 new supplementary figures and 1 new table. Changes to the main text and supplementary information, are highlighted for ease of tracking.

Response to comments by Reviewer 1

Comment 1. In the paragraph spanning lines 141 through 158, Figure 2d is referred to prior to 2c, would a rearrangement of figure be in order?

We have reordered the writing in this section so that the sub-figure references follow the order of their appearance in figure 2. (2c before 2d).

Comment 2. In the same paragraph Supplementary fig 3d is not referred to at all or anywhere else in the manuscript for that matter!

We have corrected this omission. We have now added a reference to the original Supplementary fig 3d (now supplementary Figure 4d) in the appropriate place.

Comment 3. Line 217 should read; Recombinant Satb1 robustly bound...

Corrected. Thank you.

Comment 4. Figure 3h shows the EMSA for in vitro assembled mono-nucleosomes and Satb1; these types of samples should be run on a 5% native PAGE which yields better resolution instead of an agarose gel as the authors have done. The native PAGE might reveal the 3 binding events on the three consensus Satb1 sequences better than the agarose gel. Also, the gel wells should be included in the image. Additionally, a DNA ladder as well as a free DNA (the same one that was used to reconstitute the nucleosomes) controls should be included in such gels.

While we completely agree with the reviewer that a 5% native PAGE yields better resolution than an agarose gel, native PAGE is not the ideal choice for resolving nucleo-protein complexes of mass ≥ 0.5 MDa. For larger complexes, either agarose or acrylamide-agarose composite gels have been classically used (Hellman LM and Fried MG. Nature Protocols. 2007; Ranjan et al. Cell. 2013). Since the composite mass of a Satb1 tetramer and a mono-nucleosome is ~ 0.6 MDa and for multiple binding events (e.g. 3 binding events) can result in a mass of >1 MDa, we decided to use agarose gel electrophoresis, which we have used extensively in our earlier works on analysis of nucleoprotein complexes (Nikitina et al. Science Advances, 2017).

Per reviewer's suggestion we have replaced Figure 3h with a gel which includes wells, a DNA ladder as well as a lane showing free DNA that was used to reconstitute the 601A-monomonucleosome.

Comment 5. The font size for Figure 4 parts (a-e) seems to be smaller than the rest of the figures – needs to be consistent with journal requirements

Fixed. Thank you.

Comment 6. The authors have studied the domain determinants of Satb1 binding affinity and specificity, but have not confirmed these results with their readily available in vitro nucleosomes? Did they test binding of the truncated Satb1 versions to the in vitro assembled mono-nucleosomes?

Our purpose for using in vitro EMSA was to simply assess whether Satb1 was capable of binding target sites inside the nucleosomal core unassisted by remodelers. Since reconstituted mono-nucleosomes represent a rather simplistic model of genomic chromatin, lacking in a range of important features (e.g superhelical tension that is characteristic of end-tethered chromatin, impact of shape factors of flanking DNA, periodic nature of consensus site distribution, heterochromatic vs. euchromatic state etc.) that are integral to genomic site selection by Satb1, we did not feel that it was the ideal substrate for probing the rather complex chromatin binding behavior of Satb1.

However as per reviewer's suggestion we have used EMSA to measure nucleosome binding efficiency of Δ HD, since *in vivo* binding data suggested that the Satb1 Homeodomain is dispensable for high-affinity binding to the genome. Our new data suggest that Satb1 can access nucleosome-embedded binding sites even in the absence of a functional

Homeodomain. (Supplementary Figure 15).

We have included the following section in the main text to address this issue:

“Since survival plot of track durations for FL and the different domain truncations showed that the Homeo domain is dispensable for high affinity binding, we asked whether Satb1 was capable of accessing target sites inside nucleosomal core sequences in the absence of a functional Homeodomain. Recombinant Δ HD showed efficient binding to 601A mononucleosome in an in vitro EMSA setup (Supplementary Figure 15).”

Comment 7. Line 301 has a typo- fraction has an “r” missing.

Fixed. Thank you.

Comment 8. In figure 6h, were there any R square values for the various exponential fits for survival distributions?

We have now provided the values of the rate constants of the exponential fits and the corresponding R square values in Figure 7h (originally Figure 6h) legend.

“For FL, N-C1 and Δ HD the survival distributions are fit better with two exponentials than one except for N, which exhibits mono-exponential distribution of bound molecules. The rate constants and R^2 of the fit for each construct are as follows FL (k_1 : 2.08 +/- 0.03 second⁻¹, k_2 : 0.39 +/- 0.01 second⁻¹, R^2 : 0.9994), Δ HD (k_1 : 2.12 +/- 0.06 second⁻¹, k_2 : 0.59 +/- 0.03 second⁻¹, R^2 : 0.9992), N-C1 (k_1 : 2.03 +/- 0.05 second⁻¹; k_2 : 0.5 +/- 0.02 second⁻¹, R^2 : 0.9990), N (k : 2.22 +/- 0.04 second⁻¹, R^2 : 0.9964).”

Comment 9. In line 322, I think the authors meant to refer to Figure 7e-g NOT Figure 6e-g, also in line 323, it should be Figure 7h, inset; as well as in line 326.

We thank the reviewer for pointing out the mistakes in figure references. We have now fixed these errors.

Comment 10. In Table 1, it would be easier for readers if the authors explained BF fast and slow in footnotes, so the reader doesn't need to go back into the text to recall what those stand for. Also, it would be helpful if the authors mentioned in a separate column how many cells were analyzed for each construct.

Per reviewer's suggestion we have now clearly stated that that BF stands for bound fraction in the table. We have also included the number of cells that were analyzed for each construct in a separate column for both Table 1 and the new Table 2 which summarizes the spatiotemporal FRAP results for MCF10A cell (as per suggestion by reviewer 2).

Comment 10. In the movies, it would be nice to indicate the slow and fast fractions (at least an example of each) using different colored arrows or asterisks if possible.

Per Reviewer's suggestion we have now color coded the binding events into three classes. Tracks lasting 0.5 to 1 second are marked blue, 1 to 4 seconds are marked green and greater than 4 seconds are marked red.

Response to comments by Reviewer 2

General Comment: This study provides interesting and novel insights into the targeting properties of Satb1 protein by combining various imaging and genomic approaches. In general, the experiments are well performed and the data support the conclusions.

However, the study would benefit from a more coherent analysis in the context of the T cell line and the epithelial cell line. For example, the authors use spatiotemporal FRAP and single molecule tracking of domain mutants in VL3-3M2 cells but perform ChIP-seq of domain mutants in MCF10A cells.

Per reviewer's suggestion we have now presented comparative analysis of both cell lines (MCF10A and VL3-3M2) for the following:

1. Spatiotemporal FRAP analysis of chromatin binding by different domain deletion mutants (VL3-3M2 existing data: Figure 6 and Table 1) and (MCF10A new data: New Supplementary Figure 13 and New Table 2).
2. Genome wide chromatin accessibility profile of Satb1 binding sites in MCF10A (existing data Figure 3b) and VL3-3M2 (new Supplementary Figure 7b)
3. CHIP-seq analysis of chromatin binding by different domain deletion mutants (MCF10A existing data: Figure 8) and (VL3-3M2 new data: New Supplementary Figure 18)

We could not carry out single molecule analysis of Satb1 binding in VL3 3M2 cells due to unavoidable technical limitations. To carry out spatiotemporal FRAP in VL3-3M2 cells we had to trap the thymocytes in the thermo-reversible Cygel Sustain (Biostat) to minimize drift. While Cygel lends itself to confocal imaging (FCS, Spatiotemporal FRAP), it is not suited for Hilo TIRF imaging which is at the core of all our single molecule data acquisition.

Specific comments:

Comment 1. The authors identify ~4000 Satb1-bound sites in T cells and 22000 sites in MCF10A cell in which Satb1 was ectopically expressed. The authors do not show whether the Satb1-bound sites in T cells are a subset of the Satb1-bound sites in Satb1-expressing MCF10A cells. The authors should provide an analysis of Satb1 protein levels in these cells.

We could not carry out a simple subset analysis through direct comparison of the VL3-3M2 binding sites to MCF10A, as VL3-3M2 is a mouse thymocyte cell line (which was chosen for classic Satb1 cage pattern seen in primary thymocytes) where as MCF10A is human breast epithelial cell line.

We have now provided a flow cytometry-based analysis of the protein levels in these cell lines (new Supplementary Figure 2e). We have also measured the nuclear volumes of VL3-3M2 and MCF10A cell lines (new Supplementary Figure 2a-d) to provide a realistic sense of protein concentration in these two cell lines. Comparison of Satb1 levels per unit nuclear volume in these two cell lines shows very little difference in expression (Supplementary Figure 2e).

Comment 2. It would be of interest to compare the Satb1 binding data with an analysis of gene expression in MCF10A cells without and with Satb1 expression and in T cells in which the endogenous Satb1 was knocked out and replaced by domain truncations.

While we completely agree with the reviewer that this is an interesting future direction especially in the context of Satb1 splice variants, potential changes to the transcriptional profile upon replacement of native FL Satb1 with different domain truncations may result from processes not germane to changes in binding mechanism, which is the focus of the paper. For instance, different Satb1 domains have different protein interaction modules (PDZ domain is a protein interaction hub) that can recruit coactivators/ corepressors upon binding in a context specific manner. These interactions, which go beyond the basic mechanism of binding, will have substantial impact on transcription based on domain specific interaction partners as well secondary regulators.

Comment 3a. In Figure 3, the authors show that ~95% of the Satb1 peaks in Satb1-expressing MCF10A cells are associated with transposase-inaccessible regions. Are these sites also inaccessible in MCF10A cells? This could be visualized as a heat map, similar to Figure 3C.

We have included a Venn diagram of Satb1 binding site accessibility in native MCF10A cells without any exogenous Satb1 expression (new Supplementary Figure 7a). Only 4% of Satb1 binding sites fall in accessible regions in native MCF10A cells which is very similar to MCF10A cells expressing Satb1. We have included the following in the main text.

"In sharp contrast, the bulk of the Satb1 target sites fell in transposase-inaccessible regions native MCF10A cells (~96%) lacking Satb1 expression (Supplementary Figure 7a), in MCF10A cells stably expressing Satb1 (~95%) (Figure 3b) as well as native VL3-3M2 cells (~95%) (Supplementary Figure 7b)."

Comment 3b. What percentage of Satb1-bound sites in T cells are associated with transposase-inaccessible regions?

About 5% of Satb1-bound sites in T cells are associated with transposase-inaccessible regions. We have included a Venn diagram of Satb1 binding site accessibility in VL3-3M2 cells (Supplementary Figure 7b).

Comment 4a. Figure 3i and 3j represent the DNA shape properties in the Satb1-bound nucleosome-free regions. This set of Satb1-bound sites represents less than 5% of Satb1-bound sites, whereas the vast majority of Satb1-bound sites is associated with transposase-inaccessible regions. Therefore, the authors should consider Figure S6 for the interpretation of the DNA shape instead of Figure 3i-k.

As the reviewers have noted, in the original draft we provided DNA shape analysis for sequences flanking all three classes of Satb1 binding sites; sites located in transposase inaccessible regions, sites located in transposase accessible regions and sites located in nucleosome free regions. This analysis revealed similar trends for the shapes of DNA flanks for all three classes of binding sites, however with clear differences in magnitude. We decided to lead with the

analysis of shape of binding sites located in nucleosome-free regions in Main text figure 3i-k (even though it represents less than 5% of all binding sites), since the shape factors for this class were least convolved with shape constraints imposed by nucleosomal histone octamer core. We have now clearly stated this in the main text:

“Although nucleosome-free regions represent the least abundant class of Satb1 binding sites, this none the less was best suited for assessing the role of DNA shape on site selectivity, since these sites are free of structural distortions imposed by the nucleosomal histone scaffold⁴⁶”

Comment 4b. The relative difference of propeller twist values between Satb1-bound and Satb1-unbound sites are comparable for the transposase-accessible and -inaccessible regions. How do the authors explain this observation?

We have now included the Nucleo-ATAC profile for Satb1 binding sites that fall exclusively in accessible regions (new Supplementary Figure 9). This shows that Satb1 binding sites even in accessible regions are primarily located inside nucleosomes which may explain the similarities in propeller twist values between Satb1-bound and unbound sites for the transposase-accessible and -inaccessible regions.

We have added the following section to the main text.

“The similarity in shape parameters for sequences flanking Satb1 binding sites in both inaccessible and accessible chromatin (Supplementary Figure 8a, b) is most likely due to nucleosome-imposed shape constraints on embedded sequence motifs and flanks. Indeed, Nucleo-ATAC profiling (Methods) of Satb1 binding sites in transposase accessible region showed preferential distribution inside nucleosomal core (Supplementary Figure 9) similar to transposase inaccessible region (Figure 3f).”

Comment 5. The analysis of Satb1 binding under enhanced torsional stress by the use of topoisomerase inhibitors is interesting. However, the authors should consider including other DNA-binding proteins, such as CTCF, in this analysis.

We appreciate that the reviewer liked the analysis of Satb1 binding under enhanced torsional stress by the use of topoisomerase inhibitors. We completely agree with the reviewer that the analysis of the role of torsional stress in nucleo-protein interactions is essential for a comprehensive understanding of binding site selectivity. However, a general use of topoisomerase inhibitors to analyze binding strength upon enhancement of torsional stress, may produce misleading results for proteins such as CTCF that preferentially bind at TAD boundaries. Recent studies have shown that while CTCF is essential for interaction between loop anchors, it is dispensable for establishment of TADs, whereas supercoiling and topoisomerases have been implicated in establishment of TADs (Barutcu et al. Nature Communications. 2018; Kubo et al. 2017 BioRxiv). The use of topoisomerase inhibitors may therefore alter CTCF binding through the perturbation of TAD boundary establishment and maintenance and not through a direct enhancement of torsional stress at CTCF binding sites.

We felt that a correlative analysis of TMP-seq signal and CHIP-seq strength at binding sites of a variety of chromatin binding proteins would be better suited to a mechanistic analysis of binding. We have now included a comprehensive analysis of TMP-seq signal vs CHIP-seq strength for a set of 10 chromatin binding proteins (new Main text Figure 5); 4 chromatin architectural proteins (Satb1, CTCF, cohesion-SA1, SMC1), 3 transcription factors with known preferences for open chromatin/ nucleosome depleted regions (Jun, JunB, Fos), BRCA1 (a sequence-nonspecific DNA binding protein that is involved in DNA break repair), Ring1b and BRD4 (an epigenetic reader that binds to acetylated histones). The details of the analysis have been included in the Methods section.

We have included the following results section in the main text.

“Since torsional stress regulates chromatin fine structure⁴⁵, we asked whether there are distinct classes of torsional states that are suited for specific chromatin-protein interactions. A genome wide correlative analysis of CHIP-seq read strength vs. TMP signal enrichment (fold enrichment over genome average, see Methods) of 10 different chromatin binding proteins revealed distinct patterns (Figure 5a-b, Methods). Fos, Jun and JunB, which are known to bind nucleosome-depleted regions (known as formaldehyde-assisted isolation of regulatory elements (FAIRE))⁴⁸ encompassing promoters and enhancers, showed strong preference for under-wound DNA. BRD449, an epigenetic reader of histone acetylation, showed no torsional bias, whereas BRCA150, a DNA damage repair protein that binds to DNA mostly in a sequence-independent manner, showed minimal torsional bias. Chromatin architectural proteins (CAPs) such as CTCF, SMC1 and cohesion-SA151 bound to sites characterized by a broader range of torsional states compared to canonical transcription factors but showed clear bias against highly under-wound DNA. Satb1 showed a visibly sharper choice for torsional states. Unlike the other CAPs, Satb1 demonstrated almost no binding to sites with TMP signal lower than the genome average and preferred slightly under-wound DNA, with binding strength decaying sharply with increase

in TMP crosslinking.”

Comment 6. Propeller twist values are relatively low for A/T-rich dinucleotides. Does it simply reflect high A/T percentage of the Satb1-bound sites (Figures S6c and 3k)?

In the original draft, we said:

“Because propeller twist strongly depends on the AT-content,⁴⁸ we compared this parameter for the Satb1-bound and unbound targets.”

We now elaborate that:

“Satb1 consensus sites (Figure 2e), located preferentially in nucleosomal core sequences, have multiple TA base steps, which are flexible and known to have a stronger propeller twisting than other relatively rigid T/A dinucleotides^{45,46}. This coupled with the frequent 10bp periodic recurrence of Satb1 consensus sites (Figure 2f) raises the intriguing possibility that the periodic enhancement of propeller twist at Satb1 binding site clusters may constitute an important selective filter. Thus, the enhanced propeller twisting in sequences flanking nucleosome-embedded consensus motif is more likely related to the unique geometry of TA base steps rather than a simple function of total A/T content.”

Comment 7. How many of the ~4000 T cell-associated Satb1-bound sites fall within the consensus motif not bound by Satb1 in MCF10A cells?

In the original draft we wrote:

“For MCF10A, 91% of all binding sites were satisfied by a single consensus motif (Figure 2e) whereas in VL3-3M2 two major motifs covered 100% (Supplementary Figure 3e) and 22% (Supplementary Figure 3f) of all binding sites, respectively.”

We have reworded this section to increase clarity.

“For MCF10A, 91% of all binding sites were satisfied by a single consensus motif (Figure 2e). 100% of all VL3-3M2-associated binding sites were satisfied by the same consensus motif (Supplementary Figure 4e) whereas 22% of these sites were also satisfied by a second closely related consensus (Supplementary Figure 4f).”

Comment 8. Figures 4a-b describe global changes on Satb1 binding on the peaks with the different read coverage. It may be helpful to extend this analysis by visualizing individual peaks as a scatterplot.

We have included the scatter plot as new Supplementary Figure 11.

Comment 9. N-C1- and Δ HD-Satb1 gain additional ~20000 occupied sites relative to full length Satb1. How do the authors explain stronger N-C1 and Δ HD binding even on the peaks that overlap with full-length Satb1 peaks (Figure 7a)? The authors should examine protein expression of Satb1 and the domain truncations.

For CHIP-seq experiments we sorted the different domain truncation cell lines using an identical fluorescence gate before proceeding with experiments. We have provided fluorescence intensity histograms of the different cell lines two days' post sorting in new Supplementary Figure 2.

One likely reason for differences in strength is differential accessibility of the GFP epitope in the different GFP-domain fusions. Alternatively, FL Satb1 could alter the epigenetic state of the sites it binds to, even upon short exposure unlike Δ HD and N-C1. Yet another possibility is that the HD harbors post translation modification sites that could fine tune Satb1 binding. We have stated these possibilities explicitly in the main text.

REVIEWERS' COMMENTS:

Reviewer #1 (Remarks to the Author):

we are satisfied with the changes and recommend this for publication.

Reviewer #2 (Remarks to the Author):

The revision improved the manuscript by including new experimental data and clarifying some questions in the text. In particular, the authors included a comprehensive analysis of TMP-seq signal versus CHIP-seq strength for multiple chromatin-binding proteins. In addition, the authors include the Nucleo-ATAC profile for Satb1-binding sites, and they assessed the relative protein expression of Satb1 and domain truncations. Together with other new data and textual changes, the manuscript is appropriate for publication.